# Discussions of Fluorescence in Selenium Chemistry: Recently Reported Probes, Particles, and a Clearer Biological Knowledge [note 1]

**DOI:** 10.3390/molecules26030692

**Published:** 2021-01-28

**Authors:** Ariq Abdillah, Prasad M. Sonawane, Donghyeon Kim, Dooronbek Mametov, Shingo Shimodaira, Yunseon Park, David G. Churchill

**Affiliations:** 1Molecular Logic Gate Laboratory, Department of Chemistry, Korea Advanced Institute of Science and Technology (KAIST), Daejeon 34141, Korea; ariqabdillah@kaist.ac.kr (A.A.); prasadsonawane@kaist.ac.kr (P.M.S.); dhk0706@kaist.ac.kr (D.K.); kg.spartak@kaist.ac.kr (D.M.); shingo@kaist.ac.kr (S.S.); yp2@illinois.edu (Y.P.); 2Center for Catalytic Hydrocarbon Functionalizations, Institute for Basic Science (IBS), Daejeon 34141, Korea; 3KAIST Institute for Health Science and Technology (KIHST) (Therapeutic Bioengineering), Daejeon 34141, Korea

**Keywords:** selenium, probe, particle, fluorescence, luminescence, phosphorescence, tellurium

## Abstract

In this review from literature appearing over about the past 5 years, we focus on selected selenide reports and related chemistry; we aimed for a digestible, relevant, review intended to be usefully interconnected within the realm of fluorescence and selenium chemistry. Tellurium is mentioned where relevant. Topics include selenium in physics and surfaces, nanoscience, sensing and fluorescence, quantum dots and nanoparticles, Au and oxide nanoparticles quantum dot based, coatings and catalyst poisons, thin film, and aspects of solar energy conversion. Chemosensing is covered, whether small molecule or nanoparticle based, relating to metal ion analytes, H_2_S, as well as analyte sulfane (biothiols—including glutathione). We cover recent reports of probing and fluorescence when they deal with redox biology aspects. Selenium in therapeutics, medicinal chemistry and skeleton cores is covered. Selenium serves as a constituent for some small molecule sensors and probes. Typically, the selenium is part of the reactive, or active site of the probe; in other cases, it is featured as the analyte, either as a reduced or oxidized form of selenium. Free radicals and ROS are also mentioned; aggregation strategies are treated in some places. Also, the relationship between reduced selenium and oxidized selenium is developed.

## 1. Introduction

Selenium has experienced an upsurge in research interest in the past 5 years, especially in the somewhat related realms of biology, sensing, and medicinal chemistry. We had originally intended for our review to involve more nanoscience, and conceptual bridging between small molecules and nanomaterials; we have made some in–roads and connections. In this review, we focus on selenium as a constituent of small molecule sensors. At the outset, aside from our general awareness, we started from a list of articles generated by an ISI search (Indexes: SCI–EXPANDED, SSCI, A&HCI, CPCI–S, CPCI–SSH, ESCI) with topical key words as follows: TOPIC: (selenide or selenium or selane) AND TOPIC: (probe or sensor) AND TOPIC: (fluoresc* or lumin*).

Typically, the selenium is part of the reactive or active site of the probe; in other cases, it is featured as the analyte, either as a reduced or oxidized form of selenium. But it seems that the Se–S bond is receiving more research attention now. In the course of reviewing, we found it important to discuss different types of reports because of a “chain link” relationship that free selenium might have with bound selenium; also the relationship that reduced selenium has with oxidized selenium are important to characterize. There are therefore closely related reports from chemists, physicists, medicinal chemists, and biologists. We also discuss selenium in physics surfaces, and nanoscience with regard to sensing and fluorescence. Quantum dots and nanoparticles for sensing, coatings and catalyst poisons are discussed. Thin films and aspects of solar conversions related to probing are included. We could not possibly go into detail regarding all 150+ papers. We focused on selected articles and show literature developments regarding selenium and molecular sensing research that is moving fast to help spread interest in the research area of organoselenium chemistry. 

We took care to include recent reports about reductants and ROS in the context of probing and fluorescence when they dealt with redox biological aspects. The fluorophore also relates to selenium in therapeutics and medicinal chemistry, and in molecular skeletons. There are important relationships between ROS, selenium, probe skeletons, artificial particles etc. The chemosensing, whether small molecule or nanoparticle in nature relate to metal ion analytes, H_2_S, sulfane, biothiols, glutathione etc. Free radicals and ROS are also mentioned; aggregation strategies were treated in the review. If the selenium is an analyte, it can range in identify from *reduced* organic selenium (selenocysteine and H_2_Se) to oxidized inorganic selenium (SeO)_n_. Reports relevant to biomolecules such as DNA and BSA are briefly mentioned as well. An introduction involves some important papers dealing with biology, disease and the environment. Lastly, a future “outlook” section helps and suggests ties between this topic. 

We hope that the product of our labor is digestible and relevant; hopefully it is found interconnected because of the proximity of the research results. Consider the results and content of these papers as building blocks for a much fuller biological understanding. A recent review by the Seferos group entitled “Photoactivity and optical applications of organic materials containing selenium and tellurium” has emerged [1]. As we prepare this review, a review mainly pertaining on selenium probes, we feel that these papers are setting a foundation for future answers. We have prepared a review on material that was extremely important to us and central in our laboratory’s research. 

### 1.1. Se in Biochemistry—New Findings Regarding Selenium in Biology and the Environment

It is interesting to consider recent breakthroughs in understanding Se in biochemistry. For example, the possible association of MnSOD Ala–9Val and GPx1 Pro198Leu (which contains 4 Se atoms) polymorphisms with disease [2]. The disorder in question is vitiligo disease, a hereditary/acquired progressive pigmentation disorder characterized by discoloration of skin as a result of melanocyte dysfunction. MnSOD Ala–9Val is an antioxidant responsible for conversion of the superoxide radical into hydrogen peroxide; GPx1 Pro198Leu is an antioxidant responsible for conversion of hydrogen peroxide into water. Polymorphisms of these enzymes are much less catalytically active. The analysis revealed no significant difference in genotype and allele frequency of MnSOD Ala–9Val and GPx198Leu polymorphisms between vitiligo patients and the non–vitiligo control group. 

There are also other important biological and environmental findings in the recent 5 years (see references [3,4,5,6,7,8,9,10,11,12,13]).

### 1.2. Biology and Disease

Chemists dealing with biological aspects and the environment are also making an impact. These include animal reproductive biology and mechanochemistry [14]. Recently, the research efforts of Alker et al. [15] explored the well-known Zinc pyr probe; it was used for determinations of free Zinc (labile Zinc). The measurements were found to be on the nanomolar level (0.09–0.42 nM); importantly, the correlations with Se concentration were also monitored. An important paper by Chen et al. [16] considers Se in the context of neurodegenerative disease; they attempt to gauge the selenium content, and use fluorescence. Mice were studied and treated with, and without, sodium selenate (Na_2_SeO_4_). 2D images (2D–DIE) were acquired and involved depleted serum proteins. Triply transgenic mice (3xTg–AD mice) were used in determinations. Selenium in biology sometimes involves discussion of specific biomolecules and organisms. Selenium is also important for animal reproduction. Piagentini et al. [17] investigated the effect of selenium in Brazilian ram sperm. From these studies, it is known from 350 days of investigation that the ram without selenium supplementation possessed a major defect in sperm morphology. The sperm has a defect in its head and the folding action of its tail due to the defect in its chromosome.

Redox signaling and antioxidant aspects of selenium appears in the literature from time to time. Some of this work accents the importance of the Se–S bond. Reports by Hamsath et al. dealing with this heteronuclear bond in particular is important for our review and timely for the community [18]. Nuthanakanti et al. designed fluorescence probes for studying the structure and conformation of the RNA decoding site (A–Site, A stands for aminoacyl) in bacteria and its interaction with certain antibiotics (aminoglycosides). The authors used modified fluorescence bases and selenium as an X-ray crystallography phasing agent (**^Se^U**) (Figure 1). While people usually use an atom such as a halogen or heavy element as a phasing agent [19], the usage of selenium has advantages. These include high intensity scattering peaks and resistance to decomposition, unlike halogens that can undergo dehalogenation if/when they are exposed to ionizing radiation such as X-rays. Because of the stacking structure, the non–bonded A–site that contains **^Se^U** has low fluorescence intensity. However, upon titration of antibiotics (e.g., paromonycin, neomycin B and tobramycin), the fluorescence intensity in solution increases progressively. Moreover, authors succeeded to observe the conformation binding of A–site with neomycin; 2–AP, which is a well–known conformation sensitive nucleoside, cannot achieve that however [20].

There have been many important and pertinent Se–related review articles produced. One of these articles is the paper entitled, “A review of bioselenol–specific fluorescent probes: Synthesis, properties, and imaging applications.” As the title states, the main stance in this article is how bioselenol action can proceed in screening of anticancer compounds. The contents involve fluorescent probes that are analyte-specific; aryl cleavable groups (2,4–dinitrophenyl–), the S–S motif as a reactive site for detection, benzoselenadiazole as a recognition group, and the Michael–type reactive receptors and groups that involve the Au–S bond as a motif are all target moeities. This report is a literature review which contains 91 references, and authored by 6 authors. The RSeH bonds are prevalent in thioredoxin reductase (TrxR), selenocysteine (Sec) and also the simple H_2_Se molecule itself. Therefore, a mixture of protein, small organic molecules and, literally, inorganic selenium itself are discussed in this article [21].

Another paper deals with polysulfides, an important set of reactive sulfur species; some of these examples emanate from discussion of “simple” garlic sulfur chemistry. An elegantly written piece bearing a wonderfully written abstract, it is inorganic in focus. In research, one can also try to track HSSeSH and Se*_n_*S_(8*−n*)_ species. The review, therefore, involves a discussion of polysulfides and their sources, as well as mixed selenosulfides. Species originate from H_2_S and SeO_3_^2−^. Nanoparticles are also covered by the discussion. Then, there is a section on their reactivity and activity. From this article, among other articles, we begin to appreciate the fuller interrelationships between ROS, RNS and RSS etc. However, it does not posit much about fluorescence or probing [6].

### 1.3. Environment

The need and importance to make analytical determinations in biology, or in the environment is quite obvious. A recent review dealing with Se in water was written by Devi et al. This review allows for various determinations to be described in one literature source. Therefore it is a good review describing field promising techniques and related to analysis of inorganic Se [22].

From a spectroscopic point of view, Dolgova et al. published an article in BBA explaining about how X–ray absorption spectroscopy (XAS) and X–ray fluorescence imaging (XFI) is used for imaging in detection of selenium in biological systems. The main advantage of these methods is that they do not need sample pre–treatment. This is convenient and applicable for studies when we want to perform biological sampling [23].

## 2. Selenium as Analyte

The presence of selenium in biology in small molecules or biomacromolecules and as a trace element makes it an important target to detect it reliably and with sensitivity in its multifarious molecular and ionic forms. Se in biology is known principally as being in the form of selenocysteine; also, the Se can possess oxidized forms.

### 2.1. Selenocysteine (Sec)

Zhang et al. reported the synthesis of an NIR fluorescence probe based on dicyanoisophorone and a 2,4–dinitrobenzenesulfonyl moiety (**Fsec**–**1**) as a detector for Sec (selenocysteine) at the emission wavelength 666 nm (Stokes shift = 195 nm). The probe posted a 138–fold enhancement (10 mM PBS buffer, pH 7.4 and 37 °C) (Figure 2). The quantum yield, upon addition of 10 equiv of Sec, was reported to be 0.46. **Fsec**–**1** shows saturation in 20 min. **Fsec** possesses a LOD as low as 1.0 nM and can be used optimally within an acid–base window of pH 5.0–9.0. **Fsec**–**1** is also photostable when irradiated with UV–Vis light for 120 min, and photostable when scanning was continued for 30 min. **Fsec**–**1** has good selectivity in the presence of biothiols, amino acids, cations, anions, selected ROS, some reducing agents, and even other selenium–containing compounds. From the MTT assay, it is known that **Fsec**–**1** has low cell toxicity in the concentration range of ≤20 mM. **Fsec**–**1** performance is evaluated in vitro by **MCF**–**7** cells and in vivo by animal models. Kunming mice and nude mice that contain tumor cells were used in this study. Is it known that **Fsec**–**1** can be used to help detect and image Sec concentration in a time–dependent manner. This was determined, either endogenously, or exogenously from a triggering agent such as Na_2_SeO_3_, dibenzyl diselenide and SeO_2_ [24].

Sec has cytoprotective effects and intracellular redox homeostasis in mitochondria and is present in important active sites (residues) in the enzyme oxidoreductase. Han et al. developed a novel fluorescent probe, **Mito–*di*NO_2_**, for the detection of Sec. The molecular structure of **Mito–*di*NO_2_** is based on heptamethine cyanine and bears two moieties (Figure 3); 2,4–dinitrobenzenesulfonamide is introduced as the responsive receptor unit. The lipophilic triphenylphosphonium cation can function as a mitochondrial targeting unit. In living cells, the probe can become accumulated in mitochondria and can selectively detect endogenous Sec concentration. The cell test of injecting carbon disulfide (CS_2_) with the probe indicated that CS_2_ exposure can induce decreased levels of Sec and lead to acute inflammation. These conditions are related to mitochondrial studies [25]. 

### 2.2. Selenium in Therapeutics, Medicinal Chemistry and Skeleton Cores

Some reports are related to molecular species that involve selenium, but are more focused in the literature on a therapeutic or a methodology angle, rather than a chemosensing one. However, a new delocalized skeleton always contributes indirectly to the chemosensing literature and the Se community [26]. The molecules are important building blocks for probes and help us think about the potential interaction of Se with these molecules. We refer the reader to Figure 3 in the article by Meng et al. which features Figure 3 which is an important reference diagram for this present discussion [27]. 

A report by Li et al. involves a family of indolozines which have been synthesized and reported; a description as to how they were prepared and their fluorescent properties were provided. This work is an excellent contribution with much “research follow–up” potential. Remarkable is that the synthetic route involves 3 different moieties that merge together. Also, the reactions, on the way to the new targets, are affecting necessary C–C, C–N, as well as C–Se bond formations (Figure 4). The breadth of the study is also important regarding the chemical derivatization. It also addresses lipophilic aspects important in derivatization studies. The ability to use RSe–SeR and then incorporate electrochemical techniques, also, was a novel and important dimension to this synthetic paper. While the novel systems were not overtly tested as fluorescent probes, they could certainly be assayed by redox trials and assessed in detail at a later time. It is not convenient to have RSeH *pre-prepared* and sitting for long times [28]. 

The recent paper by Chen et al. from 2019 involves chemical systems in which there are changes in an organic system by way of changing an atomic position. The systems are *N*–aryl acridine compounds. The parent molecule is *N*,*N*’–diphenyl dihydrodibenzo[*ac*]phenazine (DPAC) The so–called *isosteric* change from O to S to Se and also carbon in the form of [CMe_2_] makes for a chemically meaningful derivatization and discussion regarding the heavy atom effect (Figure 5). It is interesting work based on the extension to the molecular curvature that may be present in the excited state structure. One observes an increase in the emission maxima at 534, 642, and 687 nm for the O, S, and Se–containing systems, respectively. The fluorescent quantum yields (Φ_F_) too, were found to decrease in values, respectively: 0.35, 0.17, 0.02. The authors had a chance to analyze the bending angle. Also, there was an important solvent viscosity determination [29].

Regarding the synthesis of new selenium derivatives, Da Silva et al. successfully designed and synthesized a series of novel benzimidazo[1,2–a]quinoline derivatives introducing organochalcogens (S, Se) with reasonable yield involving short reaction times. The synthetic method involved transition–metal–free cascade reactions; this includes an intermolecular aromatic nucleophilic substitution (SNAr). Finally, an intramolecular Knoevenagel condensation leads to various organochalcogen–containing planar organic derivatives (Figure 6). Photophysical properties of both the sulfur and selenium derivatives show similar patterns; absorption maxima were positioned in the UV region (~355 nm), whereas the fluorescence emission was positioned in the violet–blue region (~440 nm); a relatively large Stokes shift was observed (~90 nm). The experimental fluorescence of sulfur analogues shows higher values than that of the selenium derivatives; this is commonly encountered and attributed to the intersystem crossing permitted by the selenium atom. In either the ground or excited state, this new organochalcogen derivatives did not display remarkable solvatochromism. In addition, absorption or emission maxima positions were not altered by the changing of the chalcogen. Owing to the quenching property of selenium, the fluorescence quantum yields are lower in value, in comparison to those of the sulfur analogs. SePh–functionalized derivatives are relevant to future probe design. It is a good paper on great and interesting novel compounds [30].

Some compounds containing selenium were intended for chemotherapy. Short–lived reactive oxygen species (ROS) play an essential role in physiological processes; their overproduction is associated with various diseases, including inflammatory diseases. Yu et al. reported the introduction of a novel redox–responsive theranostic micellar nanoparticles. These molecular devices can load anticancer drugs through coordination and hydrophobic interactions. By way of fluorescence, detection of the intracellular redox status is enabled. Professors Du and Li were instrumental in synthesizing and publishing polymers relevant to our discussion. The systems might be more chemically stable than other related systems that have been reported. The polymer was used to determine the intracellular redox status (Figure 7). The authors explored triple negative breast cancer (TNBC) models; these are known to be highly metastatic and invasive. The nanoparticle backbone includes an amphiphilic block copolymer loaded with a PEG segment and coumarin–based selenide; a hydrophobic polycarbonate moiety is also present. During the course of the physiological processes, due to redox stimulation, such nanoparticles show reversible alterations between selenide and selenoxides. Extreme reduction in fluorescence of the nanoparticles show a similar pattern to that of high ROS levels of cancer cells. Further studies show the dual drug–loading capability of the copolymer micelles to load cisplatin and paclitaxel and various cytotoxicity levels in TNBC cells and also normal cells (control set) were determined. In conclusion, dual drug–loaded nanoparticles were found to be much more toxic to TNBC cells, as compared to those of normal cells [31].

An enzymatic inhibitor was studied by Tang et al. for targeted breast tumor histone deacetylases (HDACs) imaging. The animal histone deacetylase 6 inhibitor, **SelSA**, was found to be selective towards HDAC6. With the Se inhibitors in hand, the researchers investigated breast tumors as a probed medium and a therapeutically targeted tissue. The authors also performed a molecular computation docking study. They obtained a western blot analysis of acetylation of alpha–tubulin; **SelSA**, as well as an estrogen receptor α (+), and TNBC tumors were tested. Treatment side effects were assessed when comparing these molecules to related inhibitors (pan–HDAC inhibitors) [32].

Recent research from Tang et al. dealt with nanoparticles bearing diselenide motifs. They synthesized **SeDSA**, an anthracene derivatives which were utilized for their photomechanism confirmed to be aggregation induced emission (AIE). Interestingly, neoplasia cells and normal cells were able to be compared in this probing study; the differences in the cytotoxicity were able to be established using these biological model systems. Hydrophobic and π–π interactions were at play in allowing aggregation of monomer (Figure 8). This research is important for those studying lipidic interactions. Such long chain molecules can possibly form micelles, or interface with systems that invoke lipid droplets. In the paper, the imaging, as well as treatment approaches, were discussed [32].

The article by Soares et al. is based on a series of organoselenotriazole compounds and the notion of mitochondrial dysfunction; the selenium–containing enzymes have selenium in the form of R–SeH groups (Figure 9). There are two main enzymes: glutathione peroxidase (GPx) and thioredoxin reductase. An in vivo study was conducted on a *Caenorhabditis elegans* worm model. Compared to the RSH “isoforms,” the present molecules have great reducing potential [33].

In relationship to H_2_S signaling, persulfides and selenosulfides and the R–SeSH motif containing compounds were importantly produced, studied and discussed by Kang et al. Mechanistic and reactivity work with Hantzsh’s ester involves a model for NADH as well. Also, a methylene blue assay was invoked. H_2_S reacts with p–aminodimethylaniline, in the presence of zinc acetate and ferric chloride to help generate methylene blue [34].

In the report from Ecker et al. cellular toxicities of three selenium–containing molecules, along with AZT (used as a reference molecule), were tested and compared. One of the molecules, **Rho 123**, was studied optically; the fluorescence data was obtained via flow cytometry; it was an AZT type of molecule. Healthy immune cells were tested; cell division was taken into account. Different stages of cell division were analyzed. They were studied in mice; also, cellular studies were prepared. The cell type was human peripheral blood mononuclear cells (PBMCs). Importantly, different aspects of the cellular responses were taken into account; apoptotic events were carefully gauged [35].

Recent article by Domracheva et al. involves very interesting systems in which Se is in the plane of the fluorophore molecule; Se is within the Pi-delocalized array of the ring-fused systems, either as a selenophenoquinolinone or selenophenocoumarin. Other related chemical and biochemical systems were also introduced and discussed. Also, the study goes into detail about necrotic, versus apoptotic cells. The process of cellular apoptosis was observed and accounted for. The activity, or rather inactivity, of the caspases was determined (caspase–7, caspase–8, and caspase–9 were studied). A key finding was that caspase–7 was found to be *activated*. One important conclusion was that the depletion of ROS leads to apoptosis, interesting because selenophenes are an important and somewhat underutilized chemical group in medicinal chemistry. This is a welcomed report regarding closely related systems that feature an in-ring selenium position and a clear biological outcome relating to an important enzyme class [36].

In terms of further derivatization strategies, a report by Cheignon et al. involves molecular labeling in which the reactive group bears isosteric changes and gives rise to a selenium species. There is a discussion of molecular tags which contain an atom of selenium. Selenium tags are those of an AVP ligand nature; also HO–Phpa–LVA ligand and the CCK–4 ligand is also known [37].

## 3. Selenium as an Active Site Constituent of Small Molecule Chemosensors

The chemosensing platforms involving selenium in a reactive position are popular and take various forms nowadays; there are a variety of such species that we wanted to include but that which we did not in deference to space provide here (See References: [38] (Figure 10), [20,39,40,41,42,43,44,45,46,47]).

The report by Soares-Paulino involves the coordination chemistry of the incipient Pd with the help of selenium binding. The reaction of the heavy metal with the selenium allows for a chemical change in the probe. The proposed *N,N*–coordination of the Se center helps complete a chelation pocket in which there are both hard and soft donor atoms involved in metal ion coordination from the ligands (Figure 11). The detection limit for the palladium(II) ion was found to be 32 nM [48].

The report by Mafireyi et al. discusses a probe in which the analyte is, in fact, a full protein. A diselenide portion of the molecule is known as the active site; the action of the probe is given in the presence of thioredoxin reductase TrxR1 (e.g., overexpressed in melanoma cancers (skin cancers). This probe is the first of its kind regarding this sensing target. The Se–nucleophile is designed to initiate attack at the carbonyl carbon (Figure 12). In this case, of importance for medicinal studies is that the binding affinity was measured as 15.9 µM. Imaging using in vitro biological results supported the superiority in analyte detection of the Se–Se-linked derivative over that of the S–S-linked derivative [49].

A report from Madibone et al. involves a BODIPY–based probe. It is functionalized at the *meso* position with a naphthyl group bearing two selenium groups in the peri positions and joined together with a single Se–Se bond (Figure 13). The design is clever in its simplicity; because of its compactness and the allowance that the two different Se atoms lie naturally asymmetrically, there is more to learn regarding which selenium is contributing how much electronically/sterically in the chemical system, and how many oxygen atoms are found to be possibly connected to the Se–Se unit when redox conditions are altered to different degrees. There is much more to explore with this fascinating system [50].

In this article from Zhang et al., the **SNARF**–**SSPy** and **SNARF**–**SeSPy** probes are put to the test for in vivo detection of hydrogen sulfide (H_2_S is an endogenous gasotransmitter); the H_2_S analyte was detected successfully by a **SNARF**–**SeSPy** probe that features selenium (Figure 14). The molecules were tested with zebra fish larvae. The **SNARF–SeSPy** was noted for its anti–interfering properties and low cellular toxicity. The **SNARF**–**SeSPy,** in particular, was shown to detect SH_2_ rapidly; the limit of detection was determined as 34 nM [51].

The report by Huang et al. describes *nanodots* known as Se–CDs; these molecules are selenium-doped by a certain percentage of carbon nanodots. The optical system was studied as a two–photon species. Free radicals such as OH**^.^** and O_2_**^.^**^−^ were studied under observation of addition of probe and cellular models (Figure 15). Mitochondria were involved; images of Se were obtained, and changes in fluorescence were observed and discussed. Oxidative stress goes to reduced ROS levels based on the implementation of fluorescence detection. The **TPP–Se–CDs** labelled Hela cells were also treated with an additional probe: MitoTracker Green. This involved a single (1–photon) monitoring MitoSox Red probe (10 µM); stimulation was enabled with PMA (phorbol 12–myristate 13–acetate). This probe bears the fluorescence pattern of 540–610 nm, and an excitation at 488 nm [52].

In a paper from Zang et al. endoplasmic reticula were targeted for a chemosensing study. Reversible detection was studied. First, with a “turn–off” fluorescence with ClO^−^, then a reversible reduction of selenium with GSH was observed (Figure 16). The by–product of GSH is likely to be the oxidized form GS–SG or GS–E. The probe has a Ph–Se containing motif. The structure is the 1,8–naphthalimide framework. The name of the probe is **ER**–**Se** and the resulting green fluorescence was quantified. ER red tracker (ER–Tracker Red) was also used. This study also reported the use of HeLa cells [53].

A report by Gonçalves et al., from 2017 reported the synthesis of *N*–(2–(butylselanyl)ethyl)pyrene–2–carboxamide from 2—butylselanyl)ethanamine and 1–pyrene carboxylic acid for Hg^2+^ detection. The authors investigated the fluorescence probe behavior in both water and acetonitrile solvents. Due to the so–called “heavy atom effect” and photoinduced electron transfer of Se, the probe has a smaller quantum yield than the 1–pyrenecarboxylic acid, either in water or acetonitrile (Figure 17). The difference of solvent polarity allows the fluorescence probe to aggregate together in water. Intermolecular π–π stacking molecular forces becomes dominant. Therefore, the UV–Vis spectrum becomes red–shifted. From the DLS study that they undertook to better understand the observed aggregation, its existence and dimension was investigated to ascertain that aggregation does occur and that the average size of an aggregate particle is 210 nm. The emission wavelength and fluorescence quantum yield (1–pyrene carboxylic acid as a standard, Φ_F_ = 0.11) in water and acetonitrile are 486 nm (Φ_F_ = 0.036), and 379 nm (Φ_F_ = 0.024), respectively (wavelength of excitation = 340 nm). The mechanism for sensing of Hg^2+^ is primarily by chelation–enhanced fluorescence quenching (CHEQ) in which the heavy atom, such as Hg, quenches the fluorescence of the ligand; this is present either in water and acetonitrile in a 1:1 complex form. However, in water, the chelation of Hg^2+^ to the fluorescence probe is preceded by the breaking up (dispersion) of the aggregate. Upon interference with other soluble metal ions, the fluorescence probe has a remarkable selectivity towards Hg^2+^ as indicated by a very low fluorescence intensity. The fluorescence probe is also sensitive. The LOD and LOQ in acetonitrile are as low as 2 μmol L^−1^ and 4 μmol L^−1^; while in water, the LOD and LOQ are 0.1 μmol L^−1^ and 0.17 μmol L^−1^, respectively. It is also applied to three samples of spring water; the complex band found at 408 nm is reliable for monitoring in real environmental matrixes [54].

### 3.1. H_2_S, Sulfane, Biothiols, Glutathione, and Anions as Analytes

A sensor involving selenium was reported in 2017 by Casuala et al., who successfully synthesized a novel selenourea moiety as a colorimetric chemosensor 1–(4–ethyl–2–oxo–2*H*–chromen–7–yl)–3–phenyl–selenourea (**L**) to help detect S^2−^ and CN^−^ selectively, in H_2_O:MeCN (75:25, *v*/*v*) through diselenide bond formation with an LOD of 1.0 × 10^−5^ M and 9.8 × 10^−6^ M, respectively (Figure 18). When S^2−^and CN^−^ are added, the 281 nm and 324 nm absorption wavelengths are lost; a new absorption wavelength at 353 nm is formed. Although it is not soluble in water, alternatively, they formed L by loading modified mesoporous nanosilica with trimethylsilyl as a hydrophobic outer side and PEG as a hydrophobic inner side. By that, L can discriminate the S^2−^ ion over CN^−^ ion by a “turn–off” fluorescence change mechanism [55].

Gao et al. designed a near–infrared (NIR) chemosensor based on an azo–BODIPY and diphenylselenol (**BD–di–SeH**) system for the detection of sulfane sulfur under hypoxic conditions. Sulfane sulfur is one of the members of the class of reactive sulfur species (RSS) (Figure 19); this class is receiving more research attention and publicity nowadays due to continued (and revitalized) research into any/all reactive and radical type species in biology. Sulfane sulfur (R–S–S–) is thought to be able to guard against cell apoptosis by reactive oxygen species (ROS) induced when the cell is under hypoxic conditions. Sulfane sulfur (R–S–S–) is suspected as the possible “true signaler” as compared to molecular H_2_S because of its propensity for reactivity. The mechanism of detection is “turn–on” fluorescence. Due to the heavy–atom effect by selenium, the fluorescence of the probe is quenched. But, when there is sulfane sulfur, the reactive intermediate **BD–DiSeSH** is formed; then, intramolecular cyclization occurs, thus allowing for the formation of azo–BODIPY. The quantum yield of **BD–DiSeH** in 10 mM HEPES, pH 7.4 and 20% fetal bovine serum was found to be 0.002 in which the absorption peak changes from 702 to 707 nm upon addition of Na_2_S_4_; fluorescence emission was found to be centered at 737 nm. **BD–DiSeH** works optimally at pH 7.4 and has an extraordinary selectivity towards sulfane sulfur in the existence of even high concentrations of other RSS. **BD–DiSeH** also has good photostability. In terms of toxicity, the MTT assay shows that **BD–DiSeH** can maintain 85% cell viability of A549 cells after they were incubated with 10 μM **BD–DiSeH** and 90% viability after they were incubated for 24 h with 5 μM **BD–DiSeH**. Therefore, the **BD–DiSeH** has low toxicity. **BD–DiSeH** can help detect various forms of sulfane sulfur, both exogenously and endogenously, in various cells such as A549, SH–SY5Y, RAW 254.7, Hela, HEK 293, HepG2, and SMMC7721. **BD–DiSeH** can also be used to observe the dynamics of sulfane sulfur concentration in hypoxic stress. Moreover, the fluorescence that **BD–DiSeH** demonstrated is also in line with the degree of hypoxic stress from the cell. **BD–DiSeH** was also evaluated in 3 dimensional multicellular spheroids (3D–MCs) in which the outer oxygen cannot go to the inner part of the cell. SH–SY5Y cells formed with the diameter of 300 μm were used as 3D–MCs. **BD–DiSeH** emits fluorescence (from the inside) and shows that **BD–DiSeH** can penetrate the 3D–MCs. Moreover, we can see the concentration of sulfane sulfur in BALB/c mice hippocampus samples. In living subjects such as BALB/c mice and zebrafish, the **BD–DiSeH** fluorescence intensity is in line with the concentration of Na_2_S_4_ added [56].

### 3.2. Free Radicals and ROS

Xie et al. designed a two–photon fluorescence system based on the coumarin skeleton and phenylselenide to achieve detection of HClO analyte at low concentrations (Figure 20). The mechanism is ratiometric; it involves oxidation of the selenide position (**Coum**–**Se**) into selenoxide, followed by syn elimination; this process makes the compound more conjugated; thus, the fluorescence emission becomes “red–shifted” (**Coum**–**NIR**). The effect of a simple Se oxidation from a divalent state to a tetravalent state providing a strong electronic and also a polarization to the molecule. The absorption wavelength and emission for **Coum**–**Se** and **Coum**–**NIR,** respectively, are 450 nm/495 nm and 580 nm/618 nm. The iso–absorption at 450 nm was chosen as the excitation wavelength for further study. With 5.0 equiv of HClO, the fluorescence increase was found to be 241–fold; the LOD for **Coum–Se** is 4.6 nM. **Coum–Se** reacts with HClO rapidly and obtained saturation in 5 s. Also, it is selective in the presence of common ROS, RNS and metal ions. The working pH range of **Coum–Se** for detecting HClO is 7–10. Furthermore, it is known that **Coum**–**NIR** also has stable fluorescence after it is incubated with other ROS and nucleophiles. From the use of an MTT assay, it is known that **Coum–Se** has low toxicity; the IC_50_ value was determined to be 155 μM. The in vitro study of **Coum–Se** is also evaluated in RAW264.7 cells at various concentrations; an HClO stimulator such as lipopolysaccharide and phorbol myristate acetate, and an inhibitor such as 4–aminobenzoic acid hydrazide were used. *An* in vivo study was conducted on mouse liver with the condition of acute hepatitis. All of the outputted fluorescence results are in line with the fluorescence characteristics of **Coum–Se** and **Coum–NIR**. In addition, **Coum–Se** can also be used to study the effects of Cyclosporine A in relationship with kidney and liver injury in mouse subjects [57].

Kim et al. synthesized a novel Se–containing probe; the authors observed a notable π–π stacking interaction on 5–bromo–p–2–(phenylselanyl)–thiophene–substituted BODIPY upon oxidation of Se to Se=O in the solid state; the oxidation is thought to enforce π–π stacking; therefore, a hindered rotation results (clamshell form) (see papers by J. Kuret laboratory, Ohio State University [58]) and afford a change in the spectroscopic signals owing to the strongly–enforced symmetry (diamagnetic shifting) (Figure 21). The π–π stacking interaction changes the geometrical symmetry from *C_S_* to *C*_1_ with the –Se(O)–Ph plane and the BODIPY dihedral angle are nearly parallel with one another (dihedral angle 7.9°) [59].

Research by Zhao et al. involves a FRET–based species known as **CmNp**–**Sec** capable of detecting selenocysteine. A coumarin derivative and 1,8–naphthalamide are covalently linked by a morpholine linker (Figure 22). The two halves act according to an intramolecular FRET reporter. This was studied in the context of two photon absorption; optical determinations were made with **CmNp**–**Sec** as a chemosensor. The zebrafish model was used and assays with living cells were examined [60].

A recent article by Suarez et al. involves an exploration of the Se–S bond. The paper titled “Selenosulfides tethered to gem–dimethyl Esters: a robust and highly versatile framework for H_2_S probe development” involved testing of the probe in Hela cells. The core coumarin molecule with the tether can be formed as an amide, or tethered as the high affinity tether (Figure 23). Whether one form or the other is present, the molecules are interestingly meant to react with H_2_S (pKa 6.8) at the Se–S bond. We feel this is an important contribution to selenium–containing chromophore chemical research [61].

The interesting silsesquioxane–based compound was prepared and studied by Liu et al.; the probe is termed **POSS**–**Se**. It is oligomeric and polyhedral. It gives it a roughly spherical symmetry with arms extending out, evenly and equally, from all corners of the cluster. Symmetry and “roundness” aside, its goal is to allow for optical detection of the mercuric ion (Hg^2+^) (Figure 24). The conversion of each imidazole to its selenone derivatives is an interesting aspect of this research. Regarding characterization, a peak at 1678 cm^−1^ is present and assigned as a new C=S group. The limit of detection was found to be 8.5 ppb [62].

Another impactful paper by Ishii et al. involving the topic of selenium as analyte involves H_2_S detection. The partitioning between R_2_Se(O) and R_2_Se is also addressed. The authors introduce dibenzobarrelene (Dbb)– or (mono)benzobarrelene (Mbb) groups [63]. Annaka et al. has therefore successfully designed and synthesized “turn–on” fluorescence probes in order to detect H_2_S, thus allowing for a reversible selenoxide/selenide redox system to be brought into existence (Figure 25). Experimental outcomes show, **DbbSeO** and **MbbSeO** were swiftly chemically reduced to their corresponding divalent selenium version (CH_3_CN/PBS at pH 7.4) to help produce the strongly fluorescent selenide derivatives **DbbSe** and **MbbSe,** which are indirect response to analyte concentration. In comparison with other interfering biothiols such as L–cysteine (Cys) and reduced glutathione (GSH), these selenoxide analogs exhibited extreme selectivity towards H_2_S. In contrast, NaOCl quenches **DbbSe**’s strong fluorescence quickly and selectively by oxidation to afford the corresponding **DbbSeO**. The results obtained were subject to comparison to other oxidant interferents studied such as H_2_O_2_, t–BuOOH, and ONOO^−^; HOCl reveals excellent selectivity. In conclusion, **DbbSe** and **DbbSeO** are able to detect H_2_S and HOCl in a single system reversibly [64]**.**

Li et al. reported the **Fast–TRFS** probe which targets mammalian thioredoxin reductase, in particular. In this study, step by step, the investigation reveals that reduction of cyclic disulfides/diselenides by the enzyme of the ultrafast TRFS probe shows specificity along with fast responses to thioredoxin reductase (Figure 26). Furthermore, the authors carried out mechanistic studies supporting that the fluorescence signal alteration is due to reduction of the disulfide bond present; this was different from previously published mechanisms. The success of the probe relies on the ability to allow for reduction of the attached cyclic sulfides and selenides. In order to further explore **Fast–TRFS**, additional studies of response rate of cleaving cyclic disulfides/diselenides reduction by TrxR were brought under investigation. Among all TRFS systems, **Fast–TRFS** exhibits >150–fold increase in the emission intensity; maximum fluorescence intensity reached a minimum of within 1 min when incubated with TCEP, whereas the maximum fluorescence intensity was reached within 5 min when incubated with the TrxR enzyme. It was observed that **Fast–TRFS** demonstrates superior selectivity to TrxR in comparison with the **TRFS–green** and **TRFS–red** systems [65].

This report by Deshmukh et al. is an excellent new adaptation on a previous phenylselenide motif of BODIPY. The idea of chemical oxidation of reduced Se is at the center of our discussion. There are two fluorophores per each single Se center; the probe in its entirety, in fact, involves a single Se center which was found to be sensitive and selective for (potassium) superoxide (KO_2_) analytes over other analytes studied under the same conditions (Figure 27). The fluorescence responses were determined using a mammalian breast cancer cell line (MCF–7). Also, a steric issue is involved because of the manner of the phenylselenide and BODIPY conjugation. Finally, it is the PET photomechanism inhibited by the so–called “heavy atom” effect (access to ISC mechanism, and population of triplet state). The cell lines used were mammalian breast cancer cell lines. The limit of detection was found to be 4.4 µM; the “turn–on” increase was reported as 11 fold [66].

Soares–Paulino et al. reported a tellurium–rhodamine **TR** which is a small molecule probe bearing a Ph–Te–Ph unit (Figure 28). Rhodamine B was used as the basis for this soluble chemosensor. The probe was prepared in 11% overall yield. The fluorescence quantum yield was determined to be Φ_F_ = 0.41. 2.95 µM/L; ^125^Te NMR spectroscopy was also used (value = 677.2 Hz) [67].

A tellurium containing probe was studied and reported by Soares–Paulino et al.; this molecule is emissive; while perhaps not surprising, based on previous literature, the Hg^2+^ involvement is important from a health and environmental perspective. The changes in fluorescence represents chemical changes in the probe relating to the chalcogenide nature and heteroatom position (Figure 29). When in the presence of Hg^2+^, there is a distinct interaction signaled by a strong color change. The emission exists at 582 nm. It is proposed that the Te center is directly chemically oxidized by Hg^2+^ (as opposed to electrochemical oxidation). The presence of potential interference in the form of oxidant species such as H_2_O_2_, ^−^OCl, ^−^ONOO, KO_2_, as well as NO_2_^−^ were tested, reported, and discussed [68].

A report by Tian et al. involved the preparation of a novel selenadiazole derivative by a one–pot reaction. This molecule was not only able to produce three new kinds of fluorescent compounds (wavelength of emission for Hcy, Cys and GSH are 596 nm, 605 nm and 585 respectively), but also exhibit different fluorescence emissions for discriminating each biothiol (Figure 30). The cleavage of the Se–N bond in the probe is a common mechanism of reaction with biothiols; however, only Cys can produce a final product, DAAQ, through Se–N double bond cleavage. The probe goes through a one–step Se–N bond cleavage when Hcy or GSH undergoes reaction with the probe; the intermediates are more favorable for formation than DAAQ. Also, they obtained fluorescence imaging of the probe for the detection of GSH in living cells, which indicates that the probe can be used as a potential tool for distinguishing biothiols [69].

The report by Mulay et al. involved synthesis of a fluorescent probe including a selenium–containing unit used to distinguish GSH or Cys/Hcy in living systems. The –SePh group was incorporated as the responsive, reactive unit; the ester group was introduced to increase water solubility. If the probe reacts with Cys or Hcy, the amine group of the biothiols would replace the position of –SePh (Figure 31). Additionally, biothiols can undergo a secondary connection to the proximal –CHO group through intramolecular cyclization; the probe shows green fluorescence. In the case of detecting GSH, a proximal –CHO group in the system helped harvest the pendant amino group of the selfsame bound glutathione in which the –S terminus is found to displace the SePh group from the coumarin system; imine formation is confirmed by NMR data. The probe exhibited red fluorescence [70].

Mulay et al. synthesized 7–hydroxy–2,4–dimethyl–5–(phenylselanyl)isoindoline–1,3–dione (**Probe–OCl**) and 7–hydroxy–4–methyl–5–(phenylselanyl)isobenzofuran–1,3–dione (**Probe–1**) for the detection of hypochlorous anion (^−^OCl) in a lipid droplet (Figure 32). Unfortunately, **Probe 1** is not stable due to hydrolytic reaction action at the phthalate. The wavelength of the absorption, wavelength of emission, and quantum yield are 396 nm, 512 nm, and Φ_F_ = 0.032, respectively; after addition of 10 equiv of ^−^OCl, these values are 416 nm, 523 nm and Φ_F_ = 0.37 (10 mM PBS, pH 7.4). The spectrum becomes red–shifted and the quantum yield increases after the addition of ^−^OCl. The “turn–on” mechanism of detection is through PET. When ^−^OCl is added, ^−^OCl oxidizes selenide to selenoxide and ceases the quenching, thereby allowing the fluorescence to increase up to 30–fold enhancement. **Probe–OCl** is evaluated in vitro on U–2 OS and HeLa cells and can help image ^−^OCl on the lipid droplet. **Probe OCl** also has good reversibility when it is undergoing reaction with common biothiols. The LOD of probe OCl was determined to be as low as 90 nM [71].

Kim et al. in developed a fluorescent probe for the selective imaging of HOCl, **BDPP**–**DSe**, based on dipyrazolopyridine (DPP) that has been utilized as a material for blue light–emitting diodes. The probe has a molecular design in which two fluorophores are linked by a diselenide bridge (Figure 33). This strategy achieved high selectivity and sensitivity for HOCl because, as discussed elsewhere, herein electron–rich Se centers are oxidizable by HOCl. This can thus lead to dramatic fluorescence changes. **BDPP**–**DSe** exhibited strong blue emission responses for NaOCl at 436 nm (λ_ex_ = 331 nm); the other ROS tested did not show any fluorescence changes to the probe. The oxidation of diselenide in the reaction of the probe with NaOCl was confirmed by ^1^H and ^77^Se NMR spectroscopy. Therefore, the “turn–on” response can be attributed to the quenching of the photoinduced electron transfer mechanism (PET) that existed between a phenyl selenide and BDPP group. Moreover, they demonstrated that the probe can detect hypochlorite in living cells and exhibits cell membrane permeability; cytotoxicity was also assessed [72].

Cysteine hydropersulfide (Cys–SSH) plays important roles in the synthesis of sulfur–containing cofactors, cellular signaling, and modulation of enzyme activities. Han et al. in 2016 developed **Cy**–**DiSe**, a NIR fluorescent probe for the selective detection of Cys–SSH, based on a heptamethine cyanine probe skeleton. Two moieties were introduced to the probe with a different purpose; bis(2–hydroxyethyl) diselenide can be readily reduced by Cys–SSH due to its stronger nucleophilicity than that of other biothiols (Figure 34); this moiety, therefore, promotes the selectivity of the probe towards Cys–SSH; the terminal galactose residue can allow the probe to target the liver. **Cy–DiSe** exhibited a ratio fluorescence response to Cys–SSH. After the reaction of the probe with Cys–SSH, the maximum absorption wavelength changed from 790 nm to 614 nm. Cys–SSH triggered the ICT process; the probe showed a spectral blue shift. The maximum emission wavelength underwent a distinct shift from 797 nm (Φ = 0.05) to 749 nm (Φ = 0.11). The fluorescence intensity ratio (F_749_
_nm_/F_797 nm_) has a linear relationship with Cys–SSH under those concentrations measured; this indicates that the probe can be used for both quantitative and qualitative ratiometric detection of Cys–SSH. In the analyte selectivity test, the **Cy**–**DiSe** showed selective fluorescence responses to Cys–SSH when tested against other species. The probe signaling could elude interference from other species because the reactive site in the probe has weaker reactivity towards the diselenide bond, compared to the Cys–SSH system. The concentrations of endogenous Cys–SSH in living cells (HL–7702 cells, HepG2 cells, primary hepatocyte cells) was quantitatively analyzed by the fluorescence intensity ratio. The calculated concentrations of Cys–SSH were close to the results from the flow cytometry analysis and LC–MS/MS; the probe can therefore be effectively utilized in quantification of Cys–SSH in living cells. Moreover, **Cy**–**DiSe** could target the liver in vivo because the galactose residue of the probe was selectively accepted by asialoglycoprotein receptor (ASGP–R) (asialoglycoproteins are glycoproteins in which sialic residues are removed). In particular, the expression of ASGP–R is higher in tumorous liver; the probe was found to be much more accumulated in hepatic carcinoma SD rats. In conclusion, the authors successfully targeted the liver and achieved the selective detection of Cys–SSH with ratiometric imaging in vivo [73].

Su et al. reported the synthesis and fluorescence study of a novel probe based on hydroxyl coumarin (Figure 35); it is seen as being linked with two phenylselenide units through tertiary amine groups (**SC1** and **SC2**). In water and ethanol (1:1 *v*/*v*), upon the addition of Ag^+^, the maximum absorbance at 320 nm and 325 nm was found to decrease, whereas the shoulder peak at 375 and 373 nm was found to increase for **SC1** and **SC2**, respectively (Figure 36). Both of them form a 1:1 complex with Ag^+^. The emission at 445 nm (λ_exci_= 360 nm), increases up to approximately 4–fold and 4.7–fold, respectively, upon the addition of Ag^+^. Other metals that are commonly found in the environment were also tested under the same conditions but do not give significant enhancement. Interestingly, Hg^2+^ and Cu^2+^ that usually interfere with Ag^+^ exhibit lower fluorescence intensity than the others. Counter ions of Ag^+^ also do *not* significantly affect the “turn–on” fluorescence emission of the probe after interaction with the analyte. The association constant for **SC1** and **SC2**, respectively, are 1.61 × 10^8^ L mol^−1^ and 2.35 × 10^5^ L mol^−1^. **SC2** has a lower value because of the steric hindrance imposed by the methoxy group when Ag^+^ has a proclivity to bind with **SC2**. Both **SC1** and **SC2** have an optimal pH value of around 8–11 while the maximum intensity is around pH 9. Probes **SC1** and **SC2** have good analyte binding reversibility when they undergo reaction with Na_2_S and can be used to detect Ag^+^ in naturally obtained water samples [74].

Gaur et al. reported the synthesis of chalcogen–based azole and indole systems for the expressed use of staining DNA (double stranded version). Among the results, selenium–based azole (**PA5**) has superior properties among others. In PBS buffer 100 mM, pH = 7.34 shows a quantum fluorescence yield of 0.24 (Coumarin–6 in ethanol was used as a standard Φ_F_ = 0.78). Upon addition of DNA, the fluorescence intensity of **PA5** at 534 nm increases; the absorption spectrum becomes red–shifted to 456 nm (ε ≈ 22,800 M^−1^ cm^−1^). The mechanism of the fluorescence occurs through ICT. When DNA is added, electrostatic interactions between cationic **PA5** and anionic phosphate moieties of DNA occurs, hindering the intramolecular rotation in the excited state. Therefore, ICT occurs and the fluorescence of **PA5** increases. This is confirmed by DFT and circular dichroism experiments carried out using Herring sperm DNA. Nevertheless, this interaction does not change the conformation of DNA. Viscosity, pH, and ionic strength also affect fluorescence of **PA5**. In high viscosity, the molecules of **PA5** become more rigid; therefore, the fluorescence increases due to limitations in intramolecular rotation. pH and ionic strength can interfere with electrostatic interactions. In high pH and ionic strength, electrostatic interactions become weak due to electronic neutralization. The association constant of **PA5** is 7.4 × 10^3^ M^−1^, whereas its LOD is 0.48 μM. **PA5** is also more selective towards DNA than to RNA or proteins. Moreover, **PA5** has the least spillage from nucleus to cytoplasm, the least cytotoxicity (96% cell viability) and phototoxicity (>94% cell viability after 6 h exposure), and has the highest photostability either in fixed or live cells (t_1/2_ in solution = 77 min, 30% in fixed cells and 52% in live cell samples; optical loss within 20 min and 10 min irradiation, respectively) [75].

Mulay et al. synthesized a BODIPY and phenylselenide-based fluorescence probe for the detection of ^−^OCl (**1** and **2**). They investigated the heavy atom effect of substituted Cl at the 6–position on BODIPY. **Probes 1** and **2** have a maximum emission at 507 nm and 526 nm with a fluorescence intensity enhancement to 18–fold and 50–fold, respectively, by addition of 5.0 equiv NaOCl in the presence of other ROS and RNS (EtOH; 10 mM, PBS pH 7.4 1:2 *v*/*v*). With fluorescein as an external standard, the quantum yield for **Probes 1** and **2** are 0.05 and 0.01. When NaOCl is added, the quantum yield increases for both products; they were reported as Φ_F_ = 0.41 and 0.57, respectively, for **Probes 1** and **2** (Figure 37). From these results, we can observe the heavy atom effect of Cl and start to determine how Cl helps “suppress” the fluorescence of **probe 2**. From a titration with ^−^OCl, it is known that **Probe 1** and **Probe 2** possesses an LOD of 30.9 nM and 4.5 nM, respectively. Both **Probe 1** and **2** work optimally under acidic and physiological pH when they are used to detect ^−^OCl. The photomechanism at play for both **Probe 1** and **2** is photoinduced electron transfers. Cell viability is tested using **Probe 2** and WST–1 cells. **Probe 2** has low cytotoxicity. The performance of both **Probes 1** and **2** are evaluated using both MCF–7 and RAW264.7 cells. The incubated cells with ^−^OCl and H_2_O_2_ to induce other ROS show stronger fluorescence. Moreover, **Probe 2** also can be used to detect ^−^OCl triggered by lipopolysaccharide (LPS) and phorbol 12–myristate 13–acetate (PMA) when using RAW264.7 cells [45].

The squaraine dyes are an important class of fluorophore. Dicyanomethylene squaraine dyes have a 665–716 nm fluorescence emission which are intense and narrow (Figure 38). Compound **7f** in their article is of appreciable interest considering this discussion [76].

The compound is arranged to allow for cleavage of the Se–Se bond; the remaining side of the probe bearing a nucleophilic Se would then be capable of ring formation; which, in effect, helps remove the entire pendant chain; they obtained a 1,–3–oxaselenolan–2–one ring (Figure 39) [77].

Gao et al. reported a novel fluorescent probe, **Mito–SeH,** to help detect sulfane sulfur in mitochondria [78]. The probe is based on the NIR fluorophore azo–BODIPY, including a 2–hydroselenobenzoate moiety, to allow for responses with sulfane sulfur (Figure 40). Also, a lipophilic triphenylphosphonium cation moiety is introduced to the probe as a mitochondrion–targeting unit. Since the selenol group (–SeH) holds good electrophilicity, it can play a role as a strong sulfur–acceptor. If the sulfur atom in the sulfane sulfur reacts with –SeH, it would form a –Se–SH adduct and then go through a spontaneous intramolecular cyclization to free the cleavable group from the fluorophore. The probe was demonstrated to detect sulfane sulfur in mitochondria of smooth–muscle cells. It is used for evaluating the biological effects of sulfane sulfur on helping removing excess ROS concentration under hypoxic stress (hypoxia). Finally, the authors could reveal that the sulfane sulfur can protect cells from oxidative stress by inhibiting caspase–dependent apoptosis through scavenging ROS in mitochondria [78].

As reported by Jiang et al., a novel constrained and P–centered molecule was reported, also involving a terminal chalcogenide group. In this report, the authors took advantage of ^31^P and ^77^Se NMR spectroscopic characterization which made for clear characterization [75].

The article by Xu and Xian entails a morpholine tethered to the meso group of the probe BODIPY system (Figure 41). The aliphatic selenium contained in the system becomes chemically oxidized. Yet, it still participates in the PET process. The protonated form of the Se morpholine has a H^+^ on the tertiary amino position. The pKa of **BODIPY–Se** was determined to be 4.78 and suitable for the lysosome environment. After addition of H_2_O_2_, the quantum yield with the emission at 504 nm increases up to 0.51. **BODIPY–Se** also successfully allows for imaging of H_2_O_2_ in zebra fish and cancer cells [79].

## 4. Analytes Containing Selenium

The presence of selenium–containing analyte and trying to determine it selectively by chemosensing means it is an interesting and important pursuit. First, we had better appreciate the fact that ^77^Se is NMR spectroscopically active and also responsive with respect to X–ray fluorescence analysis. The nature of the analyte is an important consideration and should be seen as reduced or oxidized, or elemental in its constitution. That being said, it could actually be converted from one form to another. Aside from the research previously discussed in depth, there have also been a number of important papers that we discuss in detail below. However, naturally there are also many reports which we do not describe, but to which we simply refer. See for example references [80,81,82,83,84,85,86,87,88].

Devi et al. designed carbon quantum dots (CQDs) (Figure 42) from pyrolysis of citric acid with a nitrogen–rich supramolecular ligand (**SL**) to detect the extremely toxic *selenite* species (SeO_3_^2−^) reference. CQDs are esterified with methyl ester, followed by reaction with a Schiff base synthesized from 2–pyridinecarboxaldehyde and hydrazine hydrate to form **CQDs–SL**. The size of CQDs is around 4–5 nm whereas the quantum yield for CQDs is around 32% (standard = quinine sulfate). The maximum fluorescence intensity emission is around 500 nm (excitation wavelength = 420 nm). When reacting with SeO_3_^2−^, **CQDs–SL** fluorescence is quenched due to vibrational states that caused non–radiative recombination. With addition of 1000 ppb of SeO_3_^2−^, an 85–fold fluorescence decrease is observed. Also, the formation of a strong Se–N bond enhances the selectivity of **CQDs–SL** toward reaction with SeO_3_^−^ when compared to the other ions. The LOD and LOQ, respectively, are 0.10 and 0.003 ppb. **CQDs–SL** works optimally at pH ≥ 7.00 and can be applied to analyze SeO_3_^2−^ in real water samples with a recovery percentage of 64–116% [89].

Guan et al. designed rhodamine 6G–based fluorescence probe, 2–(2–(2–aminoethylamino)ethyl)–3′,6′–bis(ethylamino)–2′–7′–dimethylspiro[isoindoline–1,9′xanthen]–3–one (**ABDO**) (Figure 43) by undergoing reaction with rhodamine 6G with diethylenetriamine for detecting SeO_3_^2−^ in Hela cells. Green fluorescence (around 550 nm) was gradually observed in incubated HeLa cells with SeO_3_^2−^ and 2 μM **ABDO** for 210 min; this indicates that the **ABDO** can be used to detect SeO_3_^2−^ in cells. With addition of SeO_3_^2−^, the spirolactam structure changed to a conjugated **ABDO–Se** form. Furthermore, they also compare **ABDO–Se** with mitochondrial dye, *Janus Green B* and found that the **ABDSO–Se** can also be used to image the analyte in the mitochondria as well [81].

Xin et al. reported a practical ESIPT–based probe with a special covalently attached group of the form 2–(iodomethyl)benzoate. The probe **HBT**–**Se** was tested; the leaving group was designed to be selenolactone. The fluorescence emission maximum is at approximately 470 nm (Figure 44). It has a leaving group that allows for a simple ESIPT mechanism regarding the molecule that remains. The trials were used from a 0–120 µM titration. A reasonably clean isosbestic point was obtained at 317 nm. Importantly, from a practical purpose, filter papers loaded with the probe were tested and the change in color when irradiated with 365 nm irradiation was reported [81].

A paper by Guo et al. deals with the chemosensing of selenol. The probe involves Nile Blue as well as glutathione as an analyte (Figure 45). Gold nanoparticles were used; the size of the nanoparticles was 10 nm; the emission wavelength was 692 nm. Importantly, the selenocysteine detected was both endogenous as well as exogenous in nature. The LOD was determined to be 9.5 nM. The matrixes that the authors discuss refer to wider commercial purposes of foodstuffs and materials; the authors made and reported initial determinations in tea and rice [90].

Hu et al. designed a NIR fluorescence probe based on a 13 nm diameter gold nanoparticle for detecting selenol; most importantly, selenocysteine was determined in HepG2 cells. They utilize the more stable Au–Se bond to release the thiol–peptide–Cy5.5 in which gold quenches the fluorescence of Cy5.5 through a FRET mechanism (Figure 46). The excitation maximum and emission maximum of the probe are 660 and 690 nm, respectively. One nanoparticle brings about 162 ± 1 thiol–peptide–Cy5.5. In PBS buffer (0.01 M, pH 7.4, 37 °C), the fluorescence probe undergoes reaction with selenocysteine in 10 min. Moreover, the probe works optimally in a pH of 6.3–8.0. The LOD of the probe for selenocysteine is as low as 0.38 nM. The probe also exhibits selectivity towards selenocysteine in the presence of amino acids (see reference), vitamin C, thiols, and ROS; even other selenocompounds such as (CysSe)_2_, Na_2_SeO_3_ and TrxR were tested and determined to not give significant fluorescence intensity enhancement. Through the MTT assay, the incubation of HepG_2_ with a 1.0 nM probe for 24 h still affords high cell viability of the cell; this means that the probe has low cellular toxicity under the conditions tested. In the application of the probe towards HepG_2_, probes are resistant to photobleaching. The fluorescence intensity reaches a saturation point within 2 h. Also, the lower temperature allows the intracellular fluorescence intensity to decrease. This indicates that the cellular uptake involves an active process. Study with various endocytic inhibitors shows that the mechanism of cellular uptake is via endocytosis with the help from clathrin and caveolae. The probe can be used well to monitor selenol exogenously and endogenously in HepG_2_ under hypoxic conditions. The observation with the probe shows a relationship of high selenol concentration to cell apoptosis on tumor–bearing mice [91].

### Reduced Organic Selenium (Selenocysteine) (Biological)) e.g., H_2_Se

Kong et al. synthesized fluorescence probes **Hcy–H_2_Se** for H_2_Se with novel recognition moieties from 1–2–dithiane–4,5–diol. It is known that the sulfhydryl group can cleave the S–S disulfide bond in proteins and molecules (Figure 47). However, more stable and stronger 6–membered ring containing S–S bonds require stronger nucleophilic action such as H_2_Se to allow for cleavage. From that vantage point, the authors designed a fluorogenic system with such a moiety. When H_2_Se is added, the S–S bond is cleaved. Intramolecular cyclization follows and the leaving group is creating the fluorescence for the initiated hemicyanine–based dye. Fluorescence emission appears at 535 nm. In PBS buffer (10 mM, pH = 7.4), with 10 equiv H_2_Se, the probe reacts instantly; the quantum yield increases from 0.021 to 0.081. It is also known that the LOD of the probe is 6.9 × 10^−7^ M. For biotoxicity, an MTT assay protein is performed and results in the probe having low biotoxicity. For selectivity, the probe is evaluated with the interferents such as Na_2_S, selenocysteine, NAC, DTT, VC, BSA and TrxR. The probe was also evaluated with common ROS. The result shows that the probe can differentiate H_2_Se among them. The probe can also facilitate proper imaging, either under a hypoxic or normoxic environment within the HepG2 cellular environment, or in murine hepatoma H22 tumor models in which Na_2_SeO_3_ is added to the matrix. This helps demonstrate that Na_2_SeO_3_ is metabolized to H_2_Se within the cell [91]. This addition here and elsewhere with allow for some NaHSeO_3_ concentration.

Areti et al. synthesized a fluorescence probe based on dansyl linked with glucopyranosyl; in the report, the sugar is conjugated by triazole and benzenesulfonyl, nitrobenzenesulfonyl and fluorobenzenesulfonyl moieties to detecting selenocysteine. The findings are that there is a red shift when S is replaced by Se (Figure 48). L897 NPs were used. The material involves BPST and a chain molecule DSPE–PEG. Two absorption bands centered at 347 and 711 nm were identified. From their experiments, only the 4–nitrobenzenesulfonyl derivative (**^NO2^L**) gives satisfying results for the detection of free selenocysteine. Interestingly, the 2, 4–dinitrobenzenesulfonyl (**^2NO2^L**) and the pentafluorobenzenesulfonyl derivative (**^5F^L**) shows selective detection for Cys and CN^−^, respectively. In PBS buffer (pH = 7.4), **^NO2^L** shows emission enhancement at 550 nm (wavelength of excitation = 360 nm) upon the addition of selenocysteine (up to 210–fold with 10 equiv Selenocysteine). Its fluorescence is stable at a pH of 7–10. Probe **^NO2^L** was found to be selective towards selenocysteine in the presence of cysteine and 20 other amino acids. It is also selective in the presence of other thio and seleno containing compounds, except for benzylselenol that gives a 195–fold fluorescence enhancement. The LOD in PBS buffer (pH = 7.4) is obtained as low as (1.5 ± 0.2) × 10^−7^ M or 25 ± 2 ppb. The selectivity of **^NO2^L** towards selenocysteine over Cys is also evaluated by isothermal titration calorimetry. The ΔH value for selenocysteine and Cys are –382 kcal mol^−1^ and –45 kcal mol^−1^. This indicates that the reaction of **^NO2^L** and selenocysteine is thermodynamically more favorable rather than **^NO2^L** and Cys. In fetal serum bovine and coated silica gel strips, **^NO2^L** has also good sensitivity with the LOD and fluorescence enhancement: 5.0 ± 0.4 × 10^−7^ M or 80 ± 5 ppb and 175 ± 15 fold; 10 ± 1 μM or 168 ± 16 ppb and 75 ± 5 fold (540 nm), respectively. With these results in hand, it is demonstrated that **^NO2^L** is suitable for the biological environment and can be used practically in the form of silica gel strips for selenocysteine sampling. Furthermore, **^NO2^L** performance has also evaluated in HepG2 cells. **^NO2^L** can be endogenously imaged (induced by Na_2_SeO_3_) and exogenously selenocysteine with good permeability and cell viability [92].

Selenocysteine is a bioselenium compound found in the human body. Although it is found in small concentrations [~70 µg/L], its role is extremely important, especially when it is related with diseases and pathological conditions. Therefore, it is demanded of scientists for them to develop appropriate biosensors for selenocysteine. Unfortunately, the diminutive concentrations of selenocysteine and the presence of other biothiols that exist in higher concentrations, and their eager reactivity, makes this molecular detection task a challenging one. Li et al. [93] synthesized probe **1**, a fluorescence probe based on dicyanomethylene–benzopyran and 2,4–dinitrophenyl–ether that bears a near–infrared (NIR) emission and large Stokes’ shift (Figure 49). Thus, it is more biocompatible and has lower background fluorescence. When 10 μM of probe **1** is incubated with 100 μM selenocysteine in DMS–PBS buffer (10 mM, pH 7.4 1:1 (*v*/*v*), the emission band is found to become recentered from 430 nm to 560 nm; the 25–fold increase of fluorescence emission at 706 nm is observed after testing at 5 min. Kinetically, probe **1** is undergoing reaction rapidly with selenocysteine with a k_obs_ = 0.58 min^−1^. In addition, the probe is also stable at various pH values from 4–9; the fluorescence can be observed well at pH 6–9. Based on the experiment, the limit of detection from the probe is 62 nM. The selectivity of the probe is also evaluated with other biothiols, several potassium and sodium salts, metal ions and various amino acids. Probe **1** shows good selectivity and can differentiate selenocysteine despite the presence of these possible interferents. For biological environments, probe **1** is also tested in fetal bovine serum (FBS) model solution and also in HeLa cells. Over several measured times, probe **1** does not show fluorescence, whether beginning in either fetal bovine serum (FBS) or HeLa cells. But, after FBS is (i) spiked with 100 μM selenocysteine, and (ii) the HeLa cells are incubated with 5 μM (Sec)_2_, fluorescence can be observed. In addition, fluorescence is also observed when a selenocysteine–inducer like Na_2_SeO_3_, Se_2_O or dibenzyl diselenide is added to the HeLa cell. These results show that the probe is inert to the other biothiols and can be used to detect exogenous and endogenous selenocysteine. For comparison with other probes, Rhodamine B and naphthofluorescein were selected and used. While Rhodamine B possesses similar emission wavelength, and naphthofluorescein has an emission in the NIR, in terms of background fluorescence, probe **1** shows better performance than both of them [93].

A presentation of detected selenocysteine by a cyanine probe **Mito–Cy–Sec** was made by Luo et al. It was made possible by the Ar–N(Me)C(O)CHCH_2_ sidechain. The –NH_2_ free terminal group of the selenocysteine is predicted to attack the carbonyl carbon and release the carbonyl group (Figure 50). Channels 1, 2, and 3 were determined in the cell as well as by a ratiometric determinations. Different types of thyroid disease models were explored. The undertaking was to help understand more about such untreatable and possibly serious diseases. They used both Nthy–ori3–1 as a control and BHT101 cells in their assessment [93].

Zhang et al. reported **BF–1**, a probe that has a cleavable 2,4–dinitritro phenyl– group; it was tested for selenocysteine detection in tumor bearing mice models. The mice were arranged to bear MCF–7–luc tumors. The bioluminescence (like molecular luminal) ensues upon chemical cleavage of the leaving group; the bioluminescence was monitored (Figure 51). The detection limit of 8 nM; a 580–fold increase in fluorescence was observed. Low cytotoxicity was confirmed. Selenocysteine, therefore, was monitored in cancerous tissue. Different time-lengths were studied such as 1, 6, 12 h. Detection was made in vitro, in vivo, and intra–tumoral. Na_2_SeO_3_ helps in inducing selenocysteine production in biology [94].

The effect of selenenic acid was probed in living cells; the probe **NapEb** by Ungati et al. which is a blue fluorescent naphthalamide version of Ebselen which undergoes reaction with H_2_O_2_ to give the selenic acid derivative (Figure 52). The increasing H_2_O_2_ will help rupture the Se–N bond and give a fluorescence signal continued at about 475 nm. The coordination chemistry of the Se**^…^**O intramolecular interaction was also discussed in the stabilization of the oxidize forms. “Rescue” is possible with reduced glutathione. The switching of a molecule from a “turn–on” to “turn–off” was demonstrated and the probe therefore was found to be reversible [95].

Tian et al. deals with probing of H_2_Se through the use of Se from the pocket of *N*,*N*-chelation heterocycle. **Se–1** and **Se–2** would not be an example of a reversible probe in the same way as we considered probe oxidation/reduction with aryl selenide-based systems (Figure 53). It is interesting, however, to observe that we are *losing* an atom of Se, in fact, through the utilization of adding *an equivalent of Se*. The loss of Se from the pocket/heterocycle results photomechanistically in a loss of PET; therefore, the recovery of the green fluorescence intensity was achieved from the 1,8–naphthaldehyde system. Two concentration statuses were monitored and discussed: the 1% O_2_ (hypoxic state) and ~20% O_2_ (normoxic state) [96].

Dai et al. developed fluorescent probes for the detection of selenol selective probes, **Sel**–**p1** and **Sel**–**p2** (Figure 54); the species were based on boron–dibenzopyrromethene derivatives (B–BODIPY) which bears a deep red fluorophore signature. The two probes are constructed via incorporating 2,4–dinitrobenzoxy, a fluorescence *quenching* moiety, into the B–BODIPY system. Accordingly, the probes show a broad absorption range from red to NIR while they exhibit no fluorescence for **Sel**–**p1** with two 2,4–dinitrobenzene (DNB) groups and weak fluorescence for **Sel**–**p2** with one DNB. However, the probes emitted strong fluorescence at 663 nm (λ_ex_ = 650 nm) for **Sel**–**p1** and 655 nm (λ_ex_ = 650 nm) for **Sel**–**p2** when (Sec)_2_ and dithiothreitol (DTT) were added to the phosphate buffer–DMSO (*v*/*v*, 1:1) solution mixture (pH 7.4) in which the probe is present. In the selectivity test of **Sel**–**p2** toward selenocysteine, the probe exhibited a bright red fluorescence for selenocysteine and benzylselenol; for thiophenol, a weak fluorescence was observed. The proposed detection mechanism of **Sel**–**p2** toward selenocysteine was further confirmed as S_N_Ar reaction by means of ¹H NMR spectroscopy. Since the pKa value of the selenol group in selenocysteine is lower than that of biothiols, selenols exist in their selenolate forms (RSe−) under physiological conditions; thiols however exist in their neutral form (RSH). Selenolate is a stronger nucleophile than the thiolate or thiol, and therefore acts to release fluorescent BODIPY–OH via a S_N_Ar mediated reaction; therefore, such a probe has a naturally higher selectivity for selenols over thiols. Moreover, they achieved imaging of selenocysteine with **Sel**–**p2** in living cells and mouse model studies. These results show that the probe can be a promising tool for detecting and exploring biological functions of selenocysteine [97].

Under the topic of detection of selenol: Kong et al. developed a small molecule fluorescent probe, **NIR**–**H_2_Se**, for detecting endogenous H_2_Se. Selenium has anticancer properties; H_2_Se is an important metabolite involved in many physiological processes. The **NIR**–**H_2_Se** probe was synthesized based on the structural design of a near infrared emitting merocyanine dye (Figure 55); and 2,1,3–benzoselenadiazole (BS) is a trigger moiety for H_2_Se detection. The probe exhibits weak fluorescence (Φ = 0.019) in aqueous solution (10 mM PBS, pH 7.4) at 735 nm (λ_ex_ = 688 nm) due to fluorescence quenching via the heavy atom effect of selenium. However, the fluorescence intensity increases ca. 10–fold with adding the H_2_Se into the solution. In terms of the selectivity of **NIR**–**H_2_Se** for H_2_Se, the probe selectively reacts with H_2_Se over H_2_S, selenocysteine, thiols, and other molecules such as Na_2_SeO_3_, reactive oxygen species (ROS), and amino acids. The high selectivity of the probe may be achieved by the specific cleavage of Se–N bond in the BS group by H_2_Se through nucleophilic addition. In the kinetics experiment, the rapid response of the probe for H_2_Se was identified. Also, the probe showed low cytotoxicity in HepG2 cells via the MTT assay. Moreover, they observed that for Na_2_SeO_3_, the H_2_Se precursor, induced the increase of endogenous H_2_Se in the hypoxic environment, whereas H_2_O_2_ did not increase under the same conditions. This can indicate that sodium selenite induces apoptosis of tumor cells via H_2_Se metabolism, not via oxidative–stress. Subsequently, they successfully obtained in vivo fluorescence images of H22 tumor–bearing mice injected with Na_2_SeO_3_ and the fluorescent molecular probe under study. These findings imply that the probe can provide a powerful tool for investigating the biological functions of H_2_Se and the anticancer mechanisms of Se [98].

In terms of Se as analyte, Liu et al. in 2016 reported the analysis of the use of gatifloxacin (**GAT**) as a fluorescence probe for the determination of Se(IV). The researchers discovered the fluorescence quenching of **GAT** in the presence of Se(IV). They found that using acetonitrile as a solvent decreases fluorescence emission intensity the most (among other common organic solvents studied herein). Therefore, using the probe as a solvent would be beneficial to Se(IV) determinations. They used acetonitrile in water and found that a 70% acetonitrile ratio gives the highest fluorescence intensity for **GAT** (Figure 56); but the fluorescence characteristics of the **GAT**– Se(IV) solution do not obviously change from changing the volume ratio. Accordingly, the authors chose an acetonitrile volume ratio of 70%. Regarding the presence of [H^+^] and the situation with obtaining measurements at variable pH, **GAT** and **GAT**– Se(IV) solutions display a small change in fluorescence over a pH range of 2.3–10.3; but, fluorescence intensity reaches a maximum at pH 7.3. Accordingly, they used a 0.05 mol L^−1^ Tris–HCl solution with pH = 7.3 for convenience in measurement. After analyzing the effect of the reaction time and interference of common ions, they concluded that **GAT** is highly sensitive towards Se(IV); it could instantaneously give a response as it undergoes reaction with Se(IV) rapidly. After analysis of the quenched fluorescence system, the researchers estimated the detection limit to be 1.70 × 10^−6^ mol L^−1^ [99].

The research group of Zhang developed a fluorescent probe, **O–hNRSel**, for selenocysteine detection (Figure 57). The probe is based on a tetralin–xanthene fused dye; 2,4–dinitrobenzenesulfonate (DNP) is exploited to quench the fluorescence and detect selenocysteine. Due to quenching of the fluorescence by DNP, **O–hNRSel** showed almost no fluorescence; however, the intensity of fluorescence of the probe dramatically increased with the selective detection of selenocysteine. Furthermore, authors successfully used the probe for fluorescent imaging of selenocysteine in living cells with excellent biocompatibility [100].

Luan et al. reported **BPP**, a simple organic system based on biphenyl in which a S=N bond can be formed and cleaved. The so–called sulfilimine bond known in biology is invoked, discussed and tested (Figure 58). H_2_Se is able to regulate the “status” of this bond, as well as HOBr; the system is observed having a fast response to H_2_Se. SBPP was used as a model system to help examine the effects seen on stressors imposed on this S=N bond for better understanding of its existence, reactivity and targetability in biological systems. The interest in selectively breaking the reversible covalent (double) bond specifically is important and has an analogy to that of the imine bond. This is a simple yet impressive advance in redox probing in the chemosensing field. An optical change of 480 → 525 nm was determined [101].

Gao et al. discusses a fluorescent approach to monitoring the levels of different important analytes that relate to oxidative stress. Three fluorophores were used. One, a standard rhodamine 110 probe, then **NIR–H_2_Se** whose reactions with H_2_S and also dihydroethidium which reacts selectively with O_2_^−^ (Figure 59). Measurements were made under hypoxic conditions (vs. *normoxic* conditions). In the study of anticancer selenium–containing compounds, the biological conclusion ascertained by this synergetic fluorophore study was that Na_2_SeO_3_ induces cell apoptosis which is achieved under *reducing* and not oxidizing conditions [102].

Zwolak et al. reported the utility of the organic probe dichlorodihydrofluoresceine–diacetate (**DCFA–DA**) in the context of selenium chemistry (Figure 60). An organic probe was used to detect changes based on the presence ROS concentration. The triggering species used in this study is vanadate; the “cure” is selenium in the form of selenite (SeO_3_^2−^); CHO–KI cells for an in vitro study were used. The selenite concentrations of 0.5 and 1.0 µM were used. However, it was interesting that selenite was administered and trials were undertaken; results were scrutinized. As the authors designed the research tests, the selenite was not found to prevent ROS production, in this case [103].

## 5. Selenium in Physics, Surfaces, and Nanoscience with Regard to Sensing and Fluorescence

Selenium as an elemental solid was featured in a variety of reports including chemosensing studies; selenium however is most commonly encountered as the counterion (anion) in ionic materials that pair e.g., a Group 12 element with a group 16 element. For example, cadmium selenide is widely used. These species and other selenide (and telluride species) are commonly encountered and known and discussed freely in different communities such as in physics, (inorganic, physical, analytical) chemistry, nanoscience etc. There are many points relating to aspects of frontiers in science. Some issues of papers that we list here but do not comment on involve nanoparticle, CdSe materials, cancer cells, Cu(In,Ga)Se_2_ materials, zinc selenite coatings with GSH, water solubility, fluorescent and electrochemical methods, CdSe/ZnS thioglycolic acid coatings, physicochemical characterization, Beta lactoglobulin levels, O^2−^ and Cu^2+^ detection, dimercaprol–capped QD systems, hydrodynamic radii, confocal fluorescent selenous acid, etc; These and many more topics have been central in previous reports (see references [104,105,106,107,108,109,110,111,112,113,114,115,116], (Figure 61), [117,118], (Figure 62), [119] (Figure 63), [120,121,122,123,124,125,126,127,128,129]).

Karimi et al. synthesized carbon nanotube–cadmium selenide nanocomposites by electrophoretic deposition. From their investigation with the variation of 30%, 40%, 50%, 80%, and 90% also 120 V, 160 V, 200 V and 240 V potential, it is known that the optimum conditions for the synthesis of these materials is 50% of carbon nanotube; the yield of quantum dot deposition is 0.00035 g·cm^−2^ using 240 V. These results are also in line with the luminescence data in which it is found that 50% of carbon nanotube–cadmium selenide composite has the highest intensity of the others studied. In higher concentrations, the luminescence of cadmium selenide is quenched by carbon nanotubes enabled by an active electron transfer photomechanism at play. It is confirmed by SEM and TEM that the cadmium selenide forms a dense layer on the surface of the carbon nanotube [123].

Regarding selenium in therapeutics, He et al. in 2017 designed novel selenium–containing nanocomposites to help antagonize glioblastoma. The material consists of ZnS quantum dot–selenium nanoparticles loaded with anticancer RuPFP and AS1411 surface functionalization ((QDs/Se@Ru(A)). With the aptamer AS1411, QDs/Se@Ru(A) is targeting nucleolin, a protein that is usually overexpressed in cancer cells and is involved in cell proliferation and also cell growth (Figure 64). The material QDs/Se@Ru(A) has high stability in terms of the size of the particle. For 60 days in PBS, its particle size is remaining at 70–80 nm. QDs/Se@Ru(A) has a fluorescence absorbance at 591 nm. Based on their investigation, it is supported that QDs/Se@Ru(A) has *anticancer* activity at various tumor cell lines such as U87, U251, HeLa, A375, SiHa, MCF–7 and EJ cells; the material has lower toxicity in normal human cells such as Chem–5, SV, and NIH3T3 cells. In the case of glioblastoma, the main challenge is to increase the anticancer drug infiltration to the glioma cores through the blood–brain barrier. Investigations supported that QDs/Se@Ru(A) has higher permeability to the blood–brain barrier compared to RuPFP alone. From intracellular localization investigations, it is determined that actin and ATP affect the QDs/Se@Ru(A) in the process of entering the cell through endocytosis. QDs/Se@Ru(A) also showed strong green fluorescence and even enter the nucleus after 72 h. In relation to the pH scale and drug release, QDs/Se@Ru(A) release RuPFP effectively at acidic lysosomal conditions. For the mechanism of anticancer activity, QDs/Se@Ru(A) induce ROS overproduction and cell cycle arrest through a G2/M mechanism in U87 cell [130].

Waldron et al. observed the first excitonic (1–s) absorption spectrum of PbSe quantum dot (QD) polymer nanocomposite in toluene and AB9039 epoxy matrix along with the change of temperature between 0–80 °C. It is also observed that the QD is more sensitive in toluene rather than in AB9039. The performance of the QD was also evaluated by comparing it with Lumogen F Red 305 dye. The QD was found to have comparable sensitivity to detect changes in the temperature in the form of a fluorescence shift. However, the fluorescence peak of QD is having less variation due the small size of its, bearing diameters around 2.5 nm [131].

One of the limitations of quantum dots for the applications in the cell is toxicity because of the effect heavy metals have on biological systems. Vibin et al. reported that they overcame this problem by coating CdSe with silica (QD) (Figure 65). Silica coating can prevent CdSe leaching to the cell environment. Moreover, it can also increase quantum dot hydrophilicity. The absorption and emission wavelength of QDs are around 600 and 620 nm, respectively. This silica coated–CdSe quantum dot is functionalized with antibody, the epidermal growth factor receptor (EGFR), for targeting tumor cells (the probe is named **QD–Ab**). In the tumor cell, EGFR such as transferrin and folate are usually overexpressed. From their investigations, either in vitro with RADMSCs and HeLa cells, in vivo and ex vivo with mice, the **QD–Ab** probe has higher cellular internalization, fluorescence signals or higher selectivity toward tumor cells than QD does. This shows the importance of the target directing group, in this case the EFGR antibody in the form of QD for tumor detection [132].

Chinnathambi et al. investigated the biocompatability of magnetic CdSe/ZnS quantum dot micelles with the commonest blood protein (55%) human serum albumin (HSA) (3.5–5.0 g/dL). The researchers synthesized it from DSPE–PEG (2000)–biotin, CdSe/ZnS core quantum dot, Fe_3_O_4_ nanoparticle, and phospholipid solution. The biotin is used as the directing agent to the cancer cells due to such cells usually overexpressing biotin receptors such as avidin and streptavidin. CdSe/ZnS is used as the imaging agent. Thus, the cell imaging was facilitated. Fe_3_O_4_ nanoparticles are used to add magnetic properties to the micelle. The strategy is to destroy the cancer cells with the help of a magnetic field. Phospholipids are used as the structural foundation of the micelle. According to the TEM analysis, it is shown that the size of coencapsulated CdSe/ZnS and Fe_3_O_4_ nanoparticles are around 50 nm–70 nm. While the diameter of the micelle itself is around 117–265 nm with a variation ratio of CdSe/ZnS and Fe_3_O_4_. The fluorescence emission wavelength of CdSe/ZnS in the micelle is 630 nm. A study that assesses the effect of photobleaching shows that the micelle can be exposed to strong irradiation. A magnetic study using VSM shows that the CdSe/ZnS and Fe_3_O_4_ are superparamagnetic at 305 K. The authors assessed the cellular toxicity of the species, HeLa and A549 cells show good viability with the exposure of 0–25 μg mL^−1^ micelle when the cells are incubated for 24 h. The micelle interactions with HSA is through the protein’s subdomain II A. This is indicated by the changing of the peak from tryptophan214 (Trp214) located in subdomain II A and giving an absorption at 278 nm. The micelle and HSA form a ground state complex (K binding constant = 1.19 × 10^4^ M^−1^). The hydrophobic, electrostatic and hydrogen bonding interactions were found to be dominant interactions calculated by the van’t Hoff equation. The static quenching fluorescence of HSA occurs when the micelle and HSA interact. The selenocysteine and structural change of HSA occurs when they interact with the micelle. This is indicated by the increase in the α–helical content, as observed by circular dichroism spectroscopy. However, the tertiary structure still remains intact. The distance between the micelle and HSA is around 2.37 nm. Therefore, the quenching mechanism is through FRET. The micelle also shows good stability in cancer cells [124].

Due to the high nitrogen content and rich potential host–guest chemistry and derivatization chemistry of melamine, it is used frequently in industry. There are many cases of the *misuse* of melamine, however, as a dietary additive such as in milk, to help increase the apparent level of “protein” content. Singh et al. designed a mercapto–propionic acid–capped cadmium selenide quantum dot (**m–CdSe**) to help detect melamine in analytical samples (of milk, for example). From TEM analysis, the particle size of **m–CdSe** is around 3–5 nm, whereas the crystal is around 0.5 nm in diameter as determined by XRD (Figure 66). The band gap of **m–CdSe** is 2.7 eV; its absorption peak is at 400 nm. m–CdSe shows a fluorescence emission at 532 nm when it is irradiated at 360 nm (λ*_exci_*). Through a zeta potential study, it is known that the charge of **m–CdSe** is −27.2 mV, whereas that for melamine is 14.9 mV. When the two species physically interact, the zeta potential of **m–CdSe** decreases to −10.10 mV. The decreasing in zeta potential shows that **m–CdSe** and melamine which possesses opposite charge forms a complex. Hydrogen bonding and electrostatic interactions are quenching **m–CdSe** fluorescence due to the change in its surface state. **m–CdSe** reacts rapidly with melamine. This binding is indicated by the fact that the incubation time is not affecting the fluorescence emission of the **m–CdSe** fluorophore. Through a pH study, it is known that **m–CdSe** optimally works at pH 7.0. For selectivity, **m–CdSe** fluorescence is also evaluated with the presence of interference with cysteine, glycine, sucrose, tyrosine, BSA, Na_3_AsO_4_ and Pb(NO_3_)_2_. The fluorescence does not undergo a significant change when these possible interferents are present. Evaluation in real milk samples was also conducted and *m*–CdSe can detect melamine satisfactorily [110].

Rao et al. designed an Hg^2+^–based probe placed on ZnO nanorod and doped with elemental selenium and coated with 3–mercaptopropionic acid (3–MPA). The ZnO nanorods doped with selenium are sized around 50 nm and have strong orange–red fluorescence centered at 625 nm (Figure 67). Selenium was selected because it can induce fluorescence emission. The fluorescence appears because of the presence of interstitial oxygens and crystal lattice defects. The quantum yield for the ZnO nanorod doped with selenium is (Φ_F_) 0.15, as calibrated with DAPI as standard (Φ_F_ = 0.043) as. The ZnO nanorod doped with selenium that has been functionalized with 3–MPA was studied at various values of pH, temperature, and storage time; its sensitivity and selectivity also are applied to determine Hg^2+^ in real water samples. The result is that it works optimally at pH 7.0; this is due to the fact that, under acidic pH, the MPA ligand becomes protonated and released from the ZnO nanorod doped with selenium at its surface. Conversely, in the basic pH region, ZnO nanorods will enable the formation of metal hydroxide groups and it loses its functionality to detect Hg^2+^. Higher temperatures can also make 3–MPA ligands separate from ZnO nanorods doped with selenium because of the administration of large thermal energy. In storage time, even after 6 months the fluorescence emitting from this probe remains the same; this is telling of the integrity of such materials. Within the addition of Hg^2+^, the 3–MPA that quenches the fluorescence is separated from the surface of the ZnO nanorod doped with selenium. The resulting increase in fluorescence for at least 30 s was followed by the decrease of fluorescence due to the aggregation and instability of a ZnO nanorod doped with selenium. The probe has a noted extraordinary sensitivity involving a LOD of 1 pm; the selectivity for Hg^2+^ in the presence of heavy metal ion interferents, even when they are at a high concentration was noteworthy. The probe can also be used to detect Hg^2+^ in real samples in which the recovery rate was measured to be between 93–99% [111].

The research effort of Yu et al. described the fluorescence probe for the detection of selenite and selenate based on common metal oxide materials (CeO_2_, CoO, Cr_2_O_3_, Fe_2_O_3_, In_2_O_3_, Mn_2_O_3_, NiO, TiO_2_, and ZnO) and fluorescent derivatives of DNA (Figure 68). They used FAM–labeled “24 mer” DNA oligonucleotide as the fluorescence moiety. The fluorescence becomes quenched when this moiety becomes adsorbed to the metal oxide through the phosphate backbone. However, after selenite and selenate are added, displace the fluorescent DNA on the surface of the metal oxides is affected. Therefore, the fluorescence can be observed. From this investigation, a strong recovery of fluorescence is formed when selenite is added, except for the case of ZnO, NiO, CoO and TiO_2_. While, there is no significant recovery of fluorescence of/by the fluorophore when in the presence of selenite for all of the metal oxides. From an inspection of the zeta potential, it is observed that the surfaces of all metal oxides become more negative when either selenite or selenate are added. Either in media that is too much acidic (pH = 3.0) or much basic (pH =9.0), Fe_3_O_4_ can become too positively or too negatively charged. Due to excessively strong and excessively weak interactions with DNA initially present, the fluorescence recovery becomes low in both cases. With the same analogue, the Fe_3_O_4_–DNA probe also works optimally at low ionic strength (0.20 M NaCl). For analyte chemosensor applications, Fe_3_O_4_ and Fe_2_O_3_ are chosen due to their magnetic properties, whereas CeO_2_ is chosen due to its heightened fluorescence enhancement, compared to other metal oxides. The FAM–labeled C_15_ is used. In the case of Fe_3_O_4_, when selenite is added, fluorescence enhancement is observed as much as 10–fold until it becomes saturated at about a concentration of 200 μM. From the study, it is known that the K_d_ is 143 μM; the LOD was measured as 2.0 μM. The same inclination was also observed in Fe_2_O_3,_ whereas the LOD is 3.0 μM. CeO_2_ shows a stronger fluorescence intensity, up to 40.8 fluorescence units (μM^−1^), compared to 5.5 and 3.3 fluorescence units (μM^−1^) found for Fe_3_O_4_ and Fe_2_O_3_. In the case of selenite, the response is *not* significant in Fe_3_O_4_, and Fe_2_O_3_; even for CeO_2_, the response has a value of zero. Although the selectivity for selenium salts is competing with arsenic salts and phosphates, arsenite interestingly shows *lower* fluorescence than selenite does when all conditions are kept the same. Therefore, to overcome this problem, arsenate interference can be reduced first to arsenite to allow the sensor to have less interference from arsenate and to enable more selectivity for selenite. The selectivity can, therefore, be resolved by using this clever strategy [12].

Gallium selenide materials with Ce and Er doping were prepared; Ultrafast dynamics were discussed, and the topic of doping was of central importance; the GaSe, as well as GaSe:Ce 0.1% and GaSe:Er 0.1% were investigated in a systematic study. Nonlinear 1200 nm radiation and a 100 fs pulse duration were used. The three parameters reported and discussed included α_0_ (cm^−1^), β (cm/W), and ω_0_ (µm) [133].

From a report by Huang et al. entitled “Regulating the Fluorescence Emission of CdSe Quantum Dots Based on the Surface Ligand Exchange with MAA”, the authors treat the topics of fluorescence and CdSe particles that involve an introduction of a novel exchange of organic and inorganic coatings as gauged by the emission of the CdSe quantum dots. The topic is to control the emission by having surface dynamics. Studies were conducted on a porous alumina surface—it was present as a thin film. The surface dynamics involve a mercapto acetic acid MAA; ligands are present at the surface. This application would be more for semiconductors and fluorescence and emission. The emission peaks ranged from 600–900 nm (deep–trap, an energy state in the middle of the valence and conduction bands due to impurities present in the semiconductor). The process of organic–inorganic ligand exchange works by way of electron transfer; Cd–O scission and Cd–S formation were involved. Band edge emission was monitored and the maximum red shift found in this case was 28 nm [134].

Electroluminescence of ZnSe crystals was studied by Oleshko et al. regarding the ZnSe species. CVD grown ZnSe crystals, e–beam and X–ray detectors were used. ZnSe(Ga) was the most suitable material studied here. The reason is assumed by the high content of ternary complexes that consist of zinc vacancy, interstitial zinc atom and oxygen in the lattice site (VZnZnOSe). Also, additives such as gallium, tellurium, and samarium were added and these resulting materials were studied. The detection of Vavilov–Cherenkov radiation was sought. However, this radiation was not detectable with this system. The maximum was found at ~600 nm; the band edge V–C radiation exists at ~477. Pulses of ~20 ns and ~5 µs were utilized [135].

Nanoshells are a distinct and important nanomorphology. A report by Singh et al. probes the role of selenium as the CdSe material. The manuscript deals with the study of metallic nanosphere type formations in the context of CdSe particle chemistry. Also, the two–photon spectroscopic method is that which is popular with other systems, such as small molecules and in chemosensing. These CdSe emitters are exterior to the particle; the design is a metallic nanosphere covered by a dielectric shell, which is, in turn, covered by QE’s (quantum emitters). The article contains graphs showing the effect of probe on the intensity of the TP. Concentrations were tested of QDs—these are doped into SiO_2_/Au MNS [136].

Ceria nanoparticles were reported as described by Shehata et al. Different additives were selected, used, studied and otherwise detected. The plasmonic nanostructures were added to ceria; fluorescence characteristics originating from the ceria were inspected. As discussed in other places in this section, one added chemical group was the CdSe quantum dots. The usage of ceria comes from its properties such as a strong storage capacity for oxide. The CdSe material has a red shift, compared to CdS/ZnS, with everything else being the same [137].

A hairpin ssDNA–CdSeTeS quantum dot paper was reported recently. CdSeTes QD’s were explored. Cadmium selenide and tellurium sulfide, fluorescent quantum dots measurements offered that the limit of detection was on the fM level. The virus at 10^−10^ to 10^−15^ M was able to be detected by the sensor. The formation of AuNP–dsDNA–CdSeTeS was reported. The general approach, considering the virus reagent including and aside from recent research interest in COVID 19 virus detection, is a hot topic in virus research because of the current number of people affected by the 2019–2021 pandemic [138].

The role of selenium in the Mn:ZnSe material was furthered by a report by Singh et al. The hybrid nanostructure gives fluorescence in the NIR regime. Three different nanoparticle species are combined together. These are NaY_0.797_Tm_0.003_Yb_0.2_F_4_ and ZnSe (with Mn doped) and Ag components as well. Their identifications are as nanoparticles, quantum dots and nanoparticles, respectively. The utility of Se in this scenario helps create a band–to–band transition that can be optically filtered by the AGNPs and paired with the UV emission of thulium(III) (Tm^3+^:^1^G_4_ → ^3^H_6_) [139].

The work by Sadeghi and Davami deals with eutectic solvents (two e.g., solvents that comelt, or cofreeze at a temperature when held in a fixed proportion); solvents were studied with CdSe. The devices **DES**–**CdSe QDs** were studied as an ionophore chemosensor for cupric (emission 560 nm) and screened in different beverages. Selenium dioxide was used as a starting material [140].

In a report by Chang et al., the species SeO_3_^2−^ and SeO_4_^2−^ were tested in the context of gauging the HPO_3_^2−^ ability for catalyst poisoning. The tested surface was nickel oxide (NiO). The phosphine will be able to help displace the Good’s buffer (HEPES is the most effective one) and re–install an enzymatic oxidase–like activity. In this case, the resorufin production from amplex red was determined. The perspective was to consider which atomic centers, in terms of the formal oxidation state were effective in displacing the buffer: N(III), P(I), P(V), As(V), S(VI), Se(IV) Se(VI); As(III) however, was slightly active in catalyst poisoning [141].

Wang et al. reported a study of methyl mercury and mercuric (Hg^2+^) which were determined analytically in the presence of nanoparticles composed of metal chalcogenides; the physisorbed (bound) metal present on the outside is bound differently than Cd is. The chalcogenide, therefore, allows for “ore formation”. It is important to remember here as we describe this particular report, that Cd is in fact a congener (upstairs/downstairs elemental neighbor) of Hg. Photoluminescence decay curves were prepared and help reveal Hg^2+^ forms with concentrations in the range of 0–60 µM. Stern–Volmer fitting was undertaken. The increasing emission at ~600 was shown from 0–20 µM. There is a clear shift in binding energy [142].

Horstmann et al. rationally designed yellow **CdSe/ZnS–QDs** carbon quantum dots (CQDs) and investigated its effects on *Saccharomyces cerevisiae*; **CdSe/ZnS–QDs** with a stabilizing ligand as carboxylic acid and 1 % organic impurities suspended in water. Unique optical properties of quantum dots were stabilized by ZnS shell which surround the CdSe–QD core. The absorption (estimated to be 550–600 nm) and emission (570–585 nm) of the material were measured; yellow **CdSe/ZnS–QDs** sizes were found to be approximately 4.1 nm in diameter. The toxicity effects of **CdSe/ZnS–QDs** was further investigated by the authors in many ways; (a) growth assay was studied by treating **CdSe/ZnS–QDs** to *Saccharomyces cerevisiae* (S288C); (b) total RNA–extraction experiments (triplicate) were performed with the RiboPure^TM^ yeast RNA with/without exposure of **CdSe/ZnS–QDs** to *Saccharomyces cerevisiae* (S288C); (c) reactive oxygen species (ROS) detection assays were executed by culturing yeast cells which were both treated and left untreated with **CdSe/ZnS–QDs** (triplicate manner); (d) cell wall stability experiments were performed on two upregulated genes (FAF1 and SDA1) and two down–regulated genes (DAN1 and TIR1). Experimentally, it is observed that there were no negative side effects of **CdSe/ZnS–QDs** on cell viability. In comparison with non–treated cells, however, **CdSe/ZnS–QDs** (conc = 10 g mL^−1^) exhibited more sensitivity in cell wall–compromised cell study. Through further studies, no noteworthy changes in superoxide were noticed, with or without **CdSe/ZnS–QDs** treatment. Overall, results through transcriptome analysis indicated the exposure of **CdSe/ZnS–QDs** within yeast remarkably affected genes associated in multiple cellular processes [143].

Selenium as a photothermal therapy was covered in the scientific literature by the report by Zhang et al. The synthesized material incorporating 3,4–ethylenedioxythiophene (EDOT) and the selenium enclosing acceptor unit, designed as an organic D–A–D molecular dye named **SY1080** (Figure 69) (Zhang et al., 2019). The dye molecule was stated as possessing intrinsic multifunction and having many advantages such as superior fluorescent performance over other existing NIR–II fluorophores carrying 3,4–ethylenedioxythiophene (EDOT) as a backbone. Instead of the sulfur atom, the heavier selenium center was utilized to better understand its influence and effect on creating a red shifted emission (absorption 800 nm). Experimental results indicate **SY1080** absorbs optical energy, partly transformed into heat due to which cancer cell death transpires through noninvasive photothermal therapy (PTT). Furthermore, photothermal effects of **SY1080** also produce acoustic waves and are converted into photoacoustic (PA) signals to help ablate tumor cells within the context of cancer therapy. The observed excellent results with single dose and NIR laser triggering makes **SY1080**, PA/NIR–II tumor imaging, specifically PTT in dual–modal imaging [144].

Quantum dots (QDs) are extremely well–known systems generally, as well as in the field of nanotechnology and nano–bio–engineering. Specifically, manganese selenide (MnSe), for example, is attracting a lot of attention owing to its different application properties such as its low toxicity, and importance in applications that emphasize magneto–optic, spin–transport, high ionic conductivity and low electrical resistance characteristics. Deka et al. researched differently–coated rationally–designed manganese selenide (MnSe) quantum dots (QDs) by enlisting various coating agents in research, including cetyl trimethyl ammonium bromide (CTAB), thioglycolic acid (TGA) and dextran molecules. An imaging study performed by transmission electron microscopy (TEM) reveals the size of homogeneously grown QDs (between 5 and 11 nm). The most noticeable peak was found at ∼680 nm in the studies of the QDs by obtaining Raman spectra. Further studies showed a blue–violet emission prominently positioned at ∼428 nm. The experimental time–resolved PL studies indicated, *bi–exponential decay* which are characteristic, wherein the slow component (τ2) changes according to the different nature of coating agents; the fast component (τ1 ∼ 0.10 ns), however, remains nearly the same. Owing to the complete surface passivation of the QDs, the dextran–coated MnSe QDs displayed a maximum τ2 value of approximately 5.5 ns. The highly photostable and hydrophilic nature of differently coated MnSe QDs can be applied in the form of a “vibrant” probe instead of organic probes in the field of bioimaging [145].

The report from 2019 by Henthorn et al. allows us to consider further the breadth of possibilities with fluorescence. High–energy resolution fluorescence detected X–ray absorption spectroscopic (HERFD XAS) was selected and used to better understand Mo–dependent nitrogenase, the well–known bacterial enzyme. Then, the atom, once in the cluster, can rearrange to a different bridging sulfur site. The data from the “edge” analysis was used to visualize selenium within the nitrogenase cofactor. This technique confirmed that the selenium was exchanged for a sulfur in **FeMoco** [146].

It is known that highly toxic mercury occurs in organic, inorganic, and elemental forms; a sensor which was able to help “sense” all forms of mercury in biological samples below the permissible level has a number of advantages. Bala Subramaniyan and Veerappan in 2019 reported an efficient and simple method to synthesize *N*–acetyl cysteine–capped water-soluble cadmium selenide quantum dots (CdSe QDs); an emission wavelength of 554 nm was determined. Experimental studies show, in aqueous media that CdSe QDs is highly sensitive toward Hg. Furthermore, it displays high selectivity in many biological fluids such as simulated cerebrospinal fluid, saliva, urine; also, analyte detection is possible in various natural fluids such as natural juices (vegetation) from fruits and plants such as tomato, sugarcane, and lime. CdSe QDs help perform sensing of Hg by quenching of the probe fluorescence properties. An organic, inorganic and elemental mercury limit of detection (LOD) was found, as mentioned in WHO guidelines (1.62, 0.75, and 1.27 ppb, respectively). Further, the results obtained from the bright, red fluorescence staining present within E. coli cells with the help of CdSe QDs, make it an excellent biological membrane–permeable material; it is therefore appropriate for live cell imaging [147].

Regarding selenium as a sensor component, nowadays, copper–based chalcogenide (Cu_2−x_E; E = S, Se, Te; 0 ≤ x ≤ 1) nanoparticles (NPs) continue to receive tremendous attention, owing to its prospective diverse applications in various fields such as sensors, nanomedicine, biomedical devices, energy conservation and many more. Even though they are widely employed for PA imaging and phototherapy, their systematic release of copper ions and its relative effects e.g., in biological systems are still not articulated well, either by in vivo or in vitro studies. Han et al. rationally designed and introduced a Cu^2+^ responsive near–infrared (NIR) probe named **NCM**. In this material, ultrasmall copper selenide nanoparticles served as a platform for this NCM probe in making the detection of the released copper ions efficiency. This released copper ion is able to undergo reaction with non–fluorescent NCM, in particular, and converts into NCM–1; signaling involves a very strong NIR fluorescence band at 735 nm. Considering the advantages of analyte specificity. They obtained in hand, further studies were carried out to count the release of copper ions from poly-vinylpyrrolidone (PVP)–functionalized ultrasmall Cu_2−x_Se NPs in cells and living animal models; experimental outcomes reveal its features as being biodegradable and biocompatible with no major side effects found on the living tissue (mouse organs) used for confirmation when copper ion is released. RAW 264.7 murine macrophage cell studies demonstrated that copper ions from Cu_2−x_Se NPs are released quickly. Interestingly, further investigations reveal that *ultrasmall* Cu_2−x_Se NPs were allowed to be easily removed within 72 h after intravenous injection through feces and urine of animal body; no other harmful side effects were identified after the release of copper ion. These investigations give additional support to efficient applications of copper–based chalcogenide nanoparticles in medicine and nanomedicine [148].

Sun et al. designed a fluorescence probe to detect Sec based on coumarin and the 2,4–dinitrobenzenesulfonyl moiety (**3**). The authors worked to synthetically block the nitrogen atom at the 7–position twisting to enhance the fluorescence (Figure 70). Probe **3** was found to possess good water solubility; it did not need an additional co–solvent to be soluble. In 0.1 M PBS at 30 °C, the quantum yield of probe **3** is 0.0014 and after releasing the 2,4–dinitrobenzensulfonyl group it becomes 0.30 (Rhodamine B in ethanol was used as a standard = 0.88). The intensity of fluorescence of the probe at 535 nm increases upon the addition of Sec (wavelength of excitation = 380 nm). When considering the same environment, it is discovered that probe **3** has a first order reaction constant of 2.15 min^−1^ when it undergoes reaction under the presence of 10 equiv of Sec; the second order reaction constant was determined to be approximately 700 M^−1^·s^−1^. From the pH study, it was determined that probe **3** works optimally at pH 6.0; this was chosen for further study. Probe **3** has an LOD as low as 18 nm and selective towards Sec in the presence of amino acids, metal ions and biothiols. Furthermore, Probe **3** can also be used to detect Sec residues on human thioredoxin reductase (hTR) in the presence of bovine serum albumin (BSA), acetylcholinesterase (AChE), butyrylcholinesterase (BChe) and trypsin. Cytotoxicity assays were conducted on HEK293 cells and found that the probe has low toxicity. Probe **3** also evaluated to detect exogenous analyte with the trigger of Na_2_SeO_3_ on A549 cells and endogenous Sec with a normal cell line such as HEK293, COS and cancer cells such as A549, H460 and HeLa. For the exogenous study, it is found that probe **3** can image Sec well, and fluorescence is in line with the incubation time tested. While under endogenous study, probe **3** can discern between Sec concentrations found in normal cells and cancer cells; the cancer cells contain higher concentrations than normal cells. Probe **3** can also be used to image Sec in living organisms such as zebra fish [149].

## 6. Catalyst Poisons

Regarding materials in the context of catalyst poisons and biological/enzymatic poisons for a moment, the extensive study on the effect of palladium dispersion on the absorption of toxic content by various palladium–alumina sorbents was recently reported. Stepping back in time somewhat it is good to discuss the paper by Rupp et al. and references therein because it offers a different and albeit indreact perspective regarding selenium involvement with metal ions (catalyst media) The detection of heavy metals and toxic gaseous analytes also provide interesting research directions that should be followed up because of their importance. The sorbent **Pd γ–Al_2_O_3_** is studied at 5% weight percentage in synthesized fuel composed of two contaminants, Hg and H_2_S. The toxic components of interest are hydrogen selenide, arsine (AsH_3_) and mercury. The reduced sulfur, selenium and arsenic species are likely to be present under such industrial conditions. The breakthrough curves for hydrogen selenide and arsine were measured utilizing gas chromatography and ion–trap mass spectroscopy. In one study by the Rupp research group, the data obtained provided insights into the effect of Pd distribution, and also the effect of catalyst poisoning effect caused by H_2_S. The breakthrough of hydrogen selenide occurred the fastest when **Pd/Al_2_O_3_** was implemented; determined at around 40 min and in the **Pd/B/Al_2_O_3_** material at around 120 min, that while no “breakthrough” appeared for **Pd/Al_2_O_3_**. It has been reported that no “breakthrough” appears within 3 h without the presence of H_2_S. Therefore, as the results suggest, H_2_S causes sulfur poisoning and reduces the sorbent’s ability to, in turn, capture hydrogen selenide. Furthermore, SEM backscattered–electron imaging displayed that the Pd particles **on Pd/Al_2_O_3_** happened to be the smallest and most well–distributed. Because sulfur tolerance has an inverse relationship to the size of the particles**, Pd/Al_2_O_3_** is expected to have the best resistance to H_2_S poisoning. Such results confirm no “breakthrough” of hydrogen selenide within this experimental time–frame [150].

## 7. Solar Conversions Related to Probing

Flat nanoparticles were studied by Gao et al. CdSeS material was used; the target for analytical chemistry was the dimethoate (C_5_H_12_NO_3_PS_2_); this analyte allowed the stacked plate like nanodiscs to register an optical change. The CdSeS materials were made with different Se/S molar ratios (see the article for additional information). The elements of Se and S, along with Cd, could be mapped. STEm–EDX was used; also TEM measurements were obtained. The FNPL systems were prepared by a previously known method [151].

A report by Jang et al. featured Hall measurements, and involved IR imaging of the material as an important point of the work. The PbSe material is well-known. This example described is a thin film (1.35 μm thick polycrystalline form of PbSe film); a so–called chemical bath deposition (CBD) was used. The carrier type (p–type) with a carrier concentration 0.47 × 10^14^ cm^-3^ mobility (cm^2^/V × s) of 11.7 and resistivity of 3624 (ohms × cm) and material thickness of 1.5 µm were determined [152].

## 8. A Recent Report about Reductants and ROS in the Context of Probing and Fluorescence

Recent reports involving glutathione (GSH) as an analyte include a paper by Sivasankaran et al. who designed a novel green synthesis method containing carbon dots (CDs) which were used as a reduced glutathione (GSH) fluorescence probe. It was prepared by microwaving cellulose mounted on tissue paper which was passivated with ethylenediamine (EDA or *en*). The average size of the CD was 4.2 nm, whereas the absorption and emission are located at 335 and 490 nm respectively (excitation wavelength = 400 nm). From this investigation, it was found that the addition of GSH quenches the CD fluorescence through the chemical action of the amine as a Lewis base which is replaced with thiol as a Lewis acid on the surface of CDs. This replacement has to do with the electron density of CDs surface and affects its fluorescence. Furthermore, the zeta potential and hydrodynamic volume of GSH–CDs are larger than EDA–CDs indicating that GSH–CDs tend to aggregate. The quantum yield of EDA–CDs compared to fluorescein as a standard is 24%; the intensity of fluorescence is stable, starting from 5 to 18 min with the addition of GSH. The LOD and LOQ of GSH, respectively, are determined to be 1.74 × 10^−9^ M and 5.83 × 10^−8^ M [153].

A report from Li et al. involves a PET system with graphene quantum dots. This is coupled with the use of selenium nanoparticles in which the authors looked at solution cupric determinations (**af–GQDs**); this material is coupled with selenium nanoparticles in concentrations from 1 nM to 10 µM. A detection limit of 0.4 nM was determined when conditions were made optimal. Human cervical carcinoma HeLa cells and HeLa/DDP which are cis platin-resistant versions of the HeLa cells were used. A table of the linear range and comparison to other fluorescent Cu^2+^ probes were made; linear ranges were prepared and the detection limit for the probe in question was determined to be 0.4 nM [154].

An important chemosensing article deals with NIR, or far red probing, and originates from the research efforts of Qin et al. Benzothiadiazole serves as a central moiety and is flanked by four tetraphenylethylene-type groups all conveniently covalently connected as shown in Figure 71. The electronic emission was found to appear at 670 nm; the researchers studied the materials in mouse models; their blood brain vessels were subjected to analysis. In this report, the two-photon interrogation method was used; 250 and 400 µm penetration depths were achieved [155].

A nanoprobe report by Liu et al. of a probe called **GNP–Se–Casp** deals with taking advantage of the Se–Au interaction; through this coupling, allowing the caspase cascade to become interrogated. Caspases 3, 8, and 9 were detected. The interferences from GSH were found to be minimal. The cell death was captured on camera. The apoptosis of Hela was initiated by staurosporine. Real-time detection of Hela cells and casp–9 protein complex in which the probe was preincubated was achieved. There were time–course graphics of 0, 30, 45, 60, 90, 120 min. Confocal laser scanning was performed; different excitation wavelengths were used: 494, 553, and 641 nm; the corresponding emissions were determined to be 520, 580, and 662 nm (resp.). There are clear charts showing the probe, and probe with caspase, followed by probe, caspase and inhibitor. The authors underscored to the scientific community the finding that caspase is a cascade that can serve as a target for molecular probing that can be imaged [156].

An interesting report about perfluorooctanoic acid in which selenomethine was the precursor was reported by Walekar et al. Carbon quantum dots were doped with both Se and N; the importance of having both groups 15 and 16 (Periodic Table), as well as group 14 represented here stands as an interesting and important example for further research. Values of pH dependences were measured. Perfluorooctanoic acid was seen as the quenching species; therefore, it was able to be detected. Energy peaks located at 286.2 eV are assigned to C–Se or C–N, or perhaps to the C–O group. A linear plot in the Stern–Volmer plot was observed; this suggests quenching is either dynamic or static. The quantum dot device is selective towards perfluorooctanoic acid in the presence of various alcohols, carboxylic acid, perfluorooctanesulfonate, and perfluorononic acid. The photomechanism is proposed to be through CHEQ in which the formation of the complex in the excited state exists between Se and perfluororooctanoic acid which causes fluorescence quenching. The limit of detection was found to be 1.8 µM [157].

Probing of phospholipase A2 was undertaken using quantum dots by Dias et al. The enzyme known as phospholipase A2 (BaltPLA2) was extracted from *Bothrops*
*alternatus* snake venom. There are clear and important plots found in the Fourier IR spectra. CdSe and CdS were studied by circular dichroism; CD was used for the first study of such a probe involving CdSe. Myoplast cells were subjected to imaging; also, immunofluorescence of BaltPLA2 which is a recently discovered phospholipase, was studied within the culture myoblasts. DAPI (4′,6-Diamidine-2′-phenylindole dihydrochloride) was used as the nuclear stain [158].

A report by S. Ramaraj et al. deals with nanosheets. Ytterbium–doped molybdenum selenide YbMoSe_2_ which was synthesized characterized and investigated in the context of materials science. (YbMoSe_2_)MoSe_2_, important for the surface interface and with considerations with Se in materials (materials science). The chemical substitution is effected by ultrasound [159].

A report from Huang et al. involved the design of a sensor for selenium by immobilizing CdTe (PAD) quantum dots onto filter paper. Under acidic conditions, and with addition of a hydride source, H_2_Se is formed; a gas–solid reaction occurs with the immobilized CdTe resulting in the formation of CdSe which *dynamically* quenches the fluorescence (wavelength (λ) of excitation = 365 nm) of CdTe due to the escalation of incidences in which there are surface defects (Figure 72). These defects promote nonradiative electron–hole recombination. The optimum concentration for the response from 50 ng/mL Se(IV) is 2% (*m*/*v*) KBH_4_ and 10% (*v*/*v*) HCl. Ligand, analyte concentration and material size are also investigated. It was discovered that 3–MPA and the concentration of 0.4 μM gave the highest sensitivity and maximum signal. Moreover, from the size of 2–4 nm (green, yellow and red, respectively), it is found that the 4 nm species (wavelength emission around 625 nm) provided a better visual and gives greater sensitivity than others. The reaction reaches saturation after 10 min. PAD which immobilized on paper and was also selective towards Se in the presence of interferents such as cations and hydride forming elements. PAD was also durable even upon storage for 1 month at room temperature. The LOD of PAD is around 0.1 μg/L or 1.2 × 10^−9^ M with RSD (n = 7) and 2.4% better at 20 μg/L. PAD also gives good results when it is applied on four certified reference materials (hair, urine, sediment and dogfish muscle) and real sample urine [160].

Killingworth et al. have explored the use of quantum dots (QDs), particularly streptavidin conjugated with 585 nm CdSe, as probes in correlative light electron microscopy (CLEM). The crystalline core structure of QDs allows us to generate a wide range of fluorescence emission peaks based on particle size. Furthermore, the atomic weight is sufficient to yield electron density detectable by electron microscopy. They conducted correlative light– and electron microscopy (CLEM) analysis of immunolabeled ultrathin epoxy sections of human somatostatinoma tumor. Widefield fluorescence light microscopy revealed QD labeling as bright orange fluorescence forming a granular pattern within the tumor cell cytoplasm. TEM showed that QD probes attached to amorphous material contained in individual secretory granules [161].

On the topic of Se nanocrystals and quantum dots, Mir et al. reported the synthesis of CuInGaSe (CIGC) nanocrystals at room temperature by a sol–gel method which can be conducted at room temperature; the authors obtained a uniform cluster size of 75 nm. After analysis, using TEM, SEM, XRD, EDX the authors concluded the stoichiometry was Cu 0.19, In 0.24, Ga 0.76, and Se 2.7. The crystal size was 24.5 ± 0.4 nm, the cluster size was 75 ± 5 nm and the zeta–potential was found to be +8 mV. They further analyzed possibilities of water detection in D_2_O using CIGS nanocrystals. This CIGS is used to detect H_2_O that exists as a trace impurity in D_2_O. With increasing of water content, absorbance and emission spectra of CIGS nanocrystals show quenching and enhancement, respectively. Purity of D_2_O is important for nuclear reactors. The authors attributed this to the formation of complex between water and CIGS nanocrystals. The absorption and emission wavelength of CIGS, respectively, are 420 and 483 nm. The detection limits were 9 ppm for water in D_2_O and 2.5 ppm for D_2_O in water. The detection mechanism is through quenching of the absorption and enhancement of the fluorescence emission [162].

The report by Rao et al. brought a new method forward for the synthesis of ZnSe-containing materials through radiolytic reductive procedures for the detection of Cu^2+^ in solution. They used a resultant concentration of 0.25 nM ZnSO_4_ as a zinc source, 0.50 nM Na_2_SeSO_3_ as a Se source and ethylendiamine as a capping ligand. The materials were irradiated with an 18 kGy dose involving a dose rate of 4.5 kGy h^−1^. The optimal ratio of Zn and Se is from 0.5–3.0. Moreover, excessive irradiation results in the decrease of photoluminescence due to surface defects. From that method, they obtained the 5 nm size (5.6 nm thermodynamic size) of ZnSe with a 3.04 eV band gap. The mechanism of radiolytic process involves the hydrated electron (e_aq_^−^) and H**^.^** as the main strong reducing agent. Ethylenediamine is used in addition to increase solubility, it can protect Zn^2+^ reacting with e_aq_^−^. In addition, the authors also used 2–propanol as a scavenger for the strong oxidizing agent **^.^**OH to help keep e_aq_^−^ present as a main reducing agent. The mechanism of Cu^2+^ detection is through electron transfer from ZnSe to Cu^2+^; the process reduces it to Cu^+^ and a resulting “turn–off” static quenching is observed. Furthermore, it is also selective in the presence of biologically relevant metals and can detect Cu^2+^ at the nanomolar scale [163].

Chopra et al. published an article comparing the cumulative energy demand and toxicity aspects of InP and CdSe QD materials. The authors demonstrated that, although InP is less toxic than CdSe, if we look from a wider perspective, especially in terms of energy, CdSe QD are more powerful and “consume” less energy than InP QDs, either in its synthesis or performance. They emphasized the importance of the “cradle–to–grave” life cycle of quantum dots; we must also consider to decide which one is the most appropriate QD to use and can use the consideration made by Chopra et al. to heart [164].

The recent preparation of probes that detect species in the redox cycle are tangentially related to selenium biology; some reports are referenced only while some are mentioned in detail here for discussion. Analytes such as ROS or glutathione in which selenium does not expressly appear are important papers for reference. (Also, see these additional articles [165,166,167,168]).

There was the use of dichlorofluorescein diacetate (**DCF–DA**) in a report centered on GSH and GSSG analyte concentrations present with Fe(II) and Cu(II) in which their interaction with DNA is observed. In this work the change in intensity in the thio band of the IR spectrum was monitored; the peak at 2586 cm^−1^ was assigned and monitored. When oxidized, the probe **DCF–DA** conferred to a fluorescent species. Fe(II) and Cu(II) were involved in the study. DNA as a result of the action of redox active metal ions was found to be oxidatively damaged [169].

The report by Wang et al. regarding **HGc** involved testing both in vitro and in vivo. The GSH analyte adds to the side of the Cl position at the –CH=N– group (Figure 73). It undergoes a red shifting and NIR bearing chemosensing capability for glutathione. Potential interferents such as cysteine, *cystine*, methionine, HSO_3_^−^, SO_3_^2−^ and H_2_O_2_, as well as Hcy, were tested and assessed. The relative intensity in the liver was found to be the highest; the toxicity in cells was found to be low. There was a fast response and the LOD was determined to be 252 nM. Sufficient UV and fluorescence data were obtained showing a clear change in absorbance (~650 nm) and emission bands at ~720 nm. BALB/c mice were used using the study [170].

Li and coworkers reported an iridium system named **FNO2** in which peroxynitrite is detected over O_2_^−.^ and ClO^−^ (Figure 74); The emission of the probe is found in the NIR region. Response within seconds of the analytes was determined; it was designed for long luminescence lifetimes. Additional virtues of such an Ir based are time–resolved and time–resolved phospholuminescence [171].

Kim et al. reported a 7–diethylamino–3–formylcoumarin system (Figure 75). There was a reported 76 times increase in fluorescence at 480 nm; hydrazine–based detection of peroxynitrite was enabled. The reactive group is dimethyl hydrazine group –ONOO– The retention time was observed to move from 7.5 to 6.5 *min in the chromatogram [172].

## 9. Future Outlook and Suggestions

Based on this reviewed, there are some points that we list for future investigations in hopes of deepening future selenium-based research.

Relevant to reference [50], it would be important to consider mixed systems in which we have the presence of TeSe or SeTe bond.Pertaining to reference [50], the sterics of the fluorophore near the active site can be tuned, to allow for one of the two atoms to undergo reaction first, and to explore bot the kinetics and steric considerations.Relevant to [62], we seek to determine whether the structure is the same in solution. Might it have a chance to dissociate and reassociate? If so, how would that help extend the concept of such probes?Pertaining to reference [173], it would be important to consider the synthesis and study of a more expanded π–system version of this probe. In this way, we could have e.g., a NIR version of this (exact) system for use (comparison) with the same analytes and under the same conditions [141]. In terms of particles and their surfaces, there is a little information as to the inertness of SeO_3_^2−^ and SeO_4_^2−^ for example in not interfering with the so-called nanozymes.There may be much more interplay between small molecules and particles than we are currently aware of in current probing and therefore many more investigations are necessary [174].Further understanding of the fundamental chemistry/biochemistry of RSeSRs and RSeSH systems are required [18]. This will help us understand what all of their possible intracellular targets are, and to what extent they impact signaling. Besides antioxidant regeneration and peroxide radical reduction, the roles of RSeSR and RSeSHs in other systems need to be further explored.We sometimes simplistically say we cannot have ROS without the involvement of metals or irradiation. What about this state of the art holds true for RSSs? Also, that we produce ROS when we biology with light energy is important to added.There are naturally more open questions to explore [146] with the dynamics of the “belt” positions; but whether it is possible to ever go from a “belt” position to a “cubane” frame position remains to be seen. It is interesting that KSeCN can supply the Se to the cofactor; with this simple reagent, more can be discovered quickly.

## 10. Conclusions

In this relevant and up–to–date review, we lay out the research landscape with respect to a single main group element as constrained by the keywords of luminescence and probing/sensing. The literature yielded nearly 200 papers solely from the last 5 years of scientific literature (not patent literature). We had a chance to “discuss” examples in which the Se is present in the receptor. The selenium was also shown to be part of the *analyte* such as H_2_Se. We covered many systems that included in a detachable (cleavable) group in which the Se atom fully bonds, or can be a case in which the RSe detaches from or adds to an SR group in the receptor of the probe, or RSe as an exterior analyte. Various fluorophores and nanosystems were covered. Additionally, we covered recent ROS probes and glutathione probes, to evoke a broader range of ideas and create a broader perspective of recent research work. We hope this can serve as a catalyst for expediting future research.

## Figures and Tables

**Figure 1 molecules-26-00692-f001:**
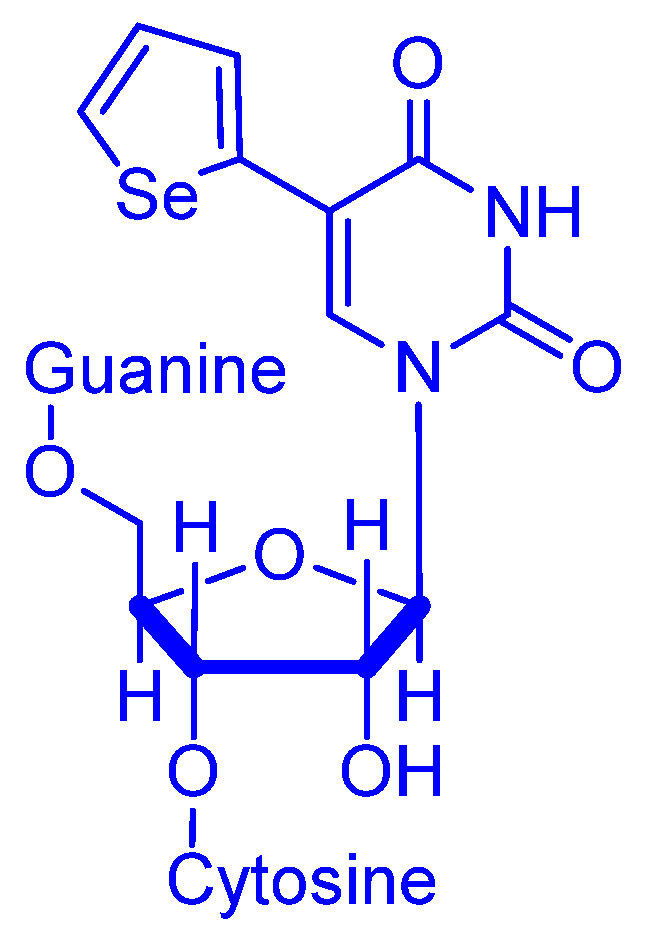
Uracil derivatization with selenophene by Nuthanakanti et al.

**Figure 2 molecules-26-00692-f002:**
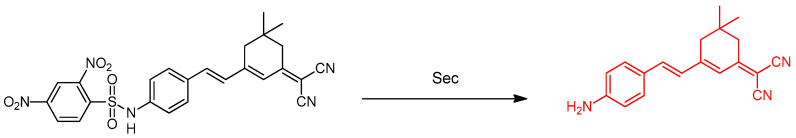
Structure of the Sec probe reported by Zhang et al. and its transformation.

**Figure 3 molecules-26-00692-f003:**
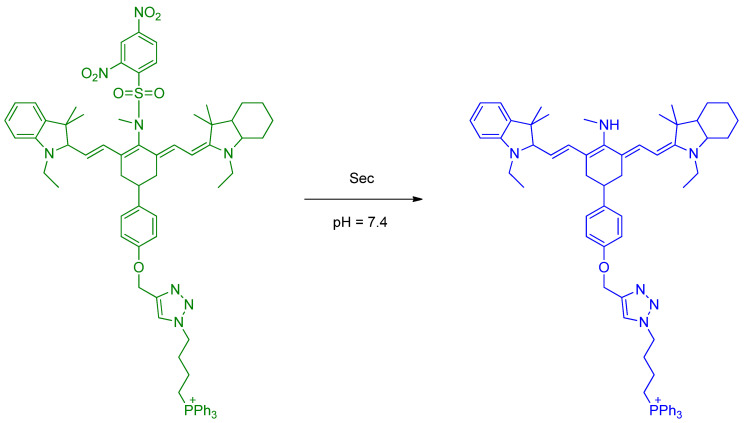
Detection mechanism of **Mito–*di*NO_2_** for Sec by Han et al.

**Figure 4 molecules-26-00692-f004:**
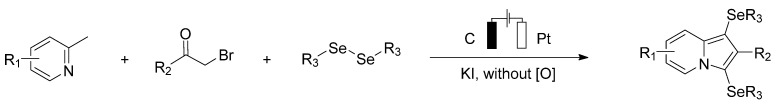
Reactions and selenium compound synthesized electrochemically by Li et al.

**Figure 5 molecules-26-00692-f005:**
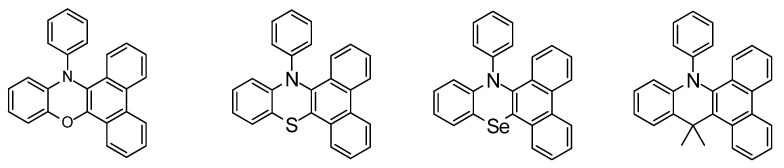
Structures of the acridine from the report by Chen et al.

**Figure 6 molecules-26-00692-f006:**
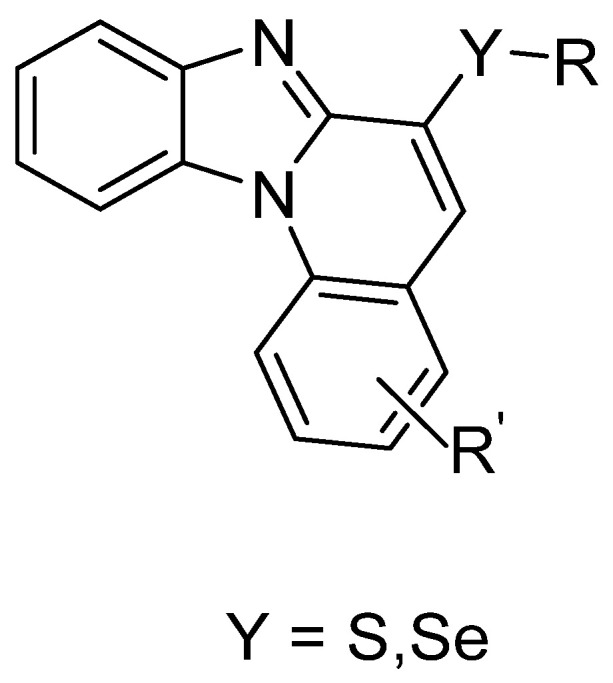
Molecular structures of the novel benzimidazo[1,2–a]quinoline derivatives reported by Da Silva et al.

**Figure 7 molecules-26-00692-f007:**
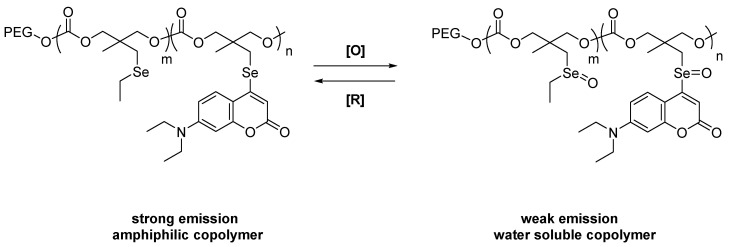
The chemical structure of the amphiphilic copolymer and the water–soluble copolymer studied in the context of nucleophilic attack and fluorescence changes by Yu et al.

**Figure 8 molecules-26-00692-f008:**
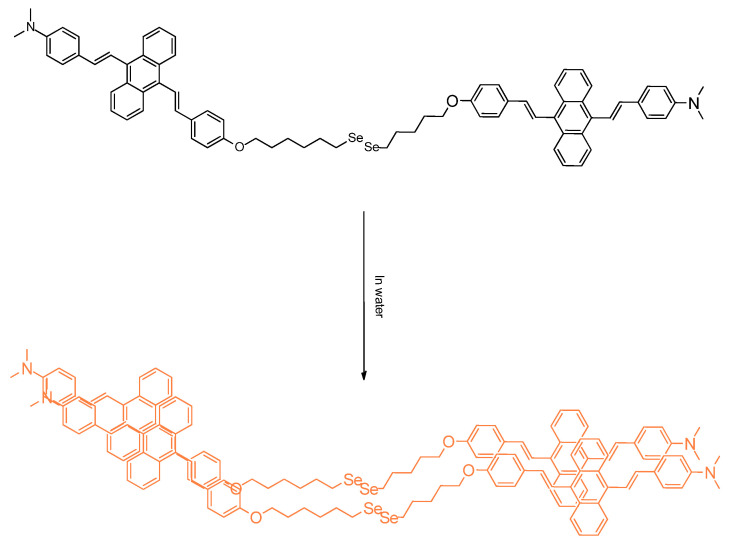
The diselenide system prepared by Tang et al. and its self–stacking form when introduced into water.

**Figure 9 molecules-26-00692-f009:**
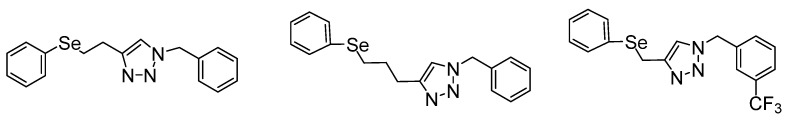
The organotriazoles studied by Soares et al.

**Figure 10 molecules-26-00692-f010:**
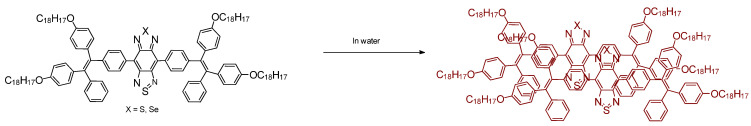
The compound and its aggregation tendency in water was described by Wu et al. The position of the selenium is specific and discussed.

**Figure 11 molecules-26-00692-f011:**
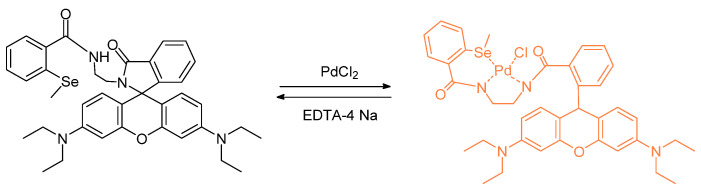
Chemical structure and reactivity of the rhodamine–containing selenide (**Rh**–**Se**) system designed for Pd^2+^ binding and recognition by Soares–Paulino et al.

**Figure 12 molecules-26-00692-f012:**
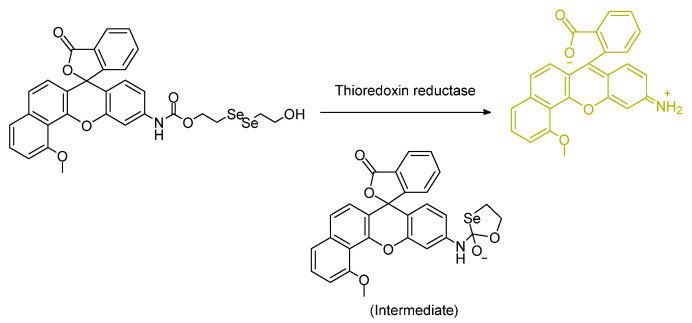
The first TrxR diselenide–based probe reported by Mafireyi, et al.

**Figure 13 molecules-26-00692-f013:**
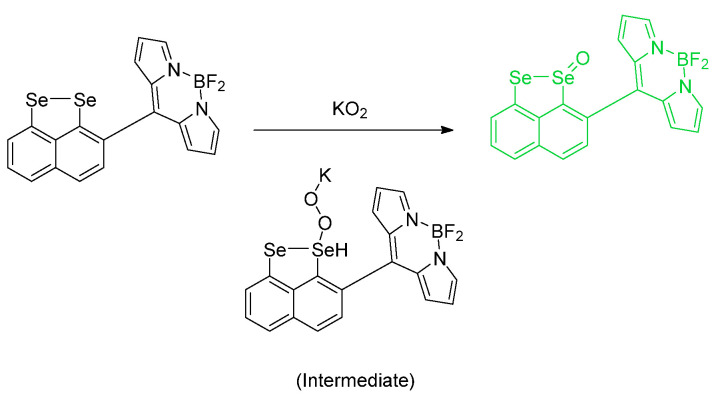
The recent novel diselenide probe reported by Madibone et al.

**Figure 14 molecules-26-00692-f014:**
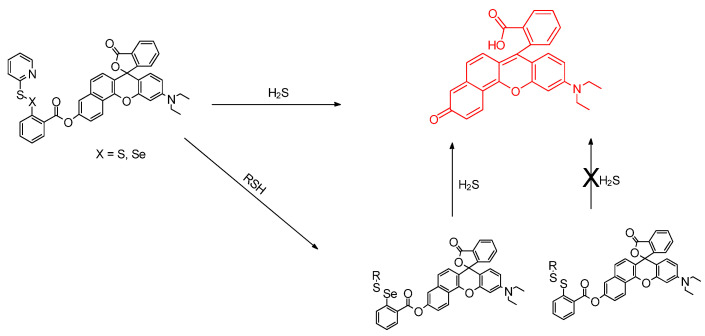
Image of **SNARF**–**SSPy** and **SNARF**–**SeSPy** by Zhang et al.

**Figure 15 molecules-26-00692-f015:**
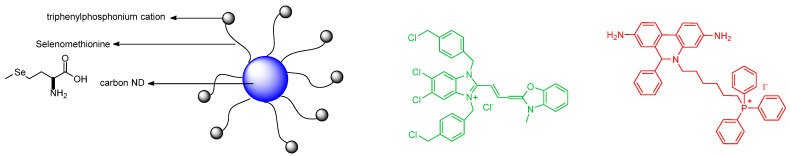
Structures of **TPP–Se–CDs** by Huang et al. and MitoTracker Green and MitoSox Red.

**Figure 16 molecules-26-00692-f016:**
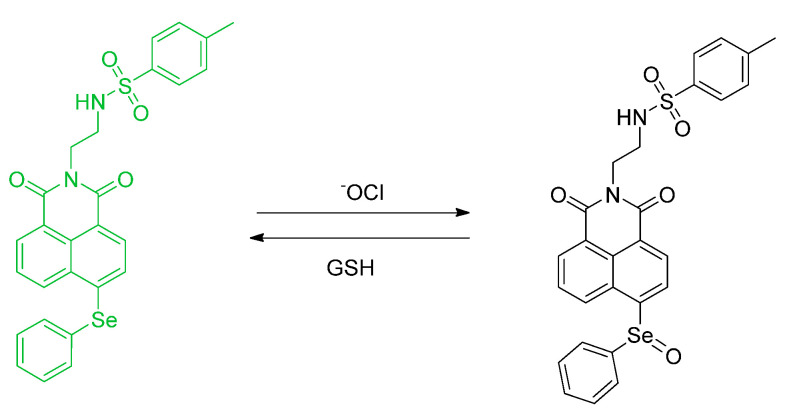
The structure and reactivity of the reversible/reserrable Se–Ph containing species reported by Zang et al.

**Figure 17 molecules-26-00692-f017:**
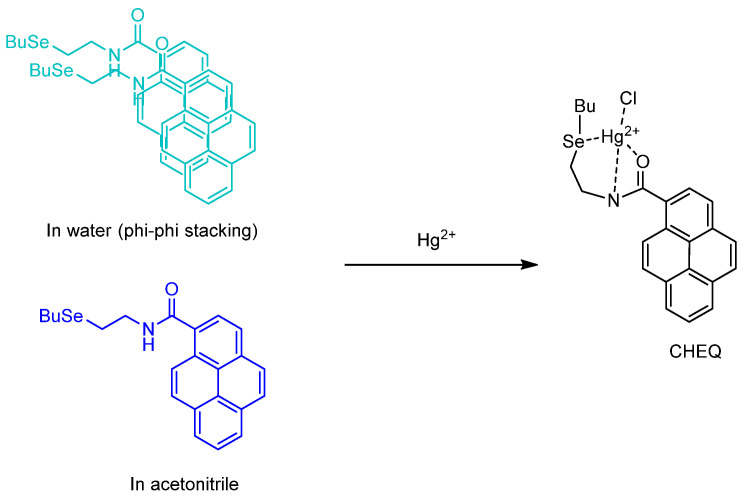
The selenium-containing probe by Gonçalves et al. that engages in π–π stacking and binds to mercury.

**Figure 18 molecules-26-00692-f018:**
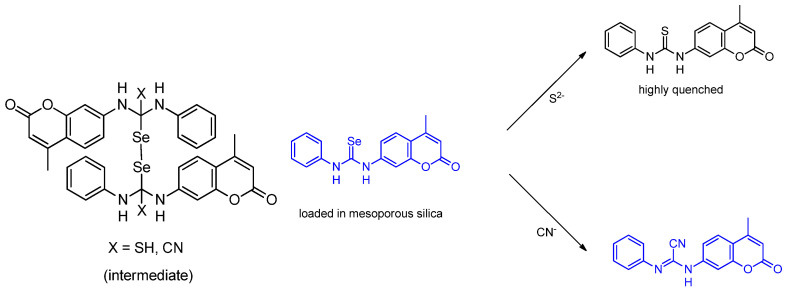
The structure of chemosensor 1–(4–ethyl–2–oxo–2*H*–chromen–7–yl)–3–phenyl–selenourea (L) which, as synthesized by Casula et al. demonstrated, the detection of S^2−^ and CN^−^.

**Figure 19 molecules-26-00692-f019:**
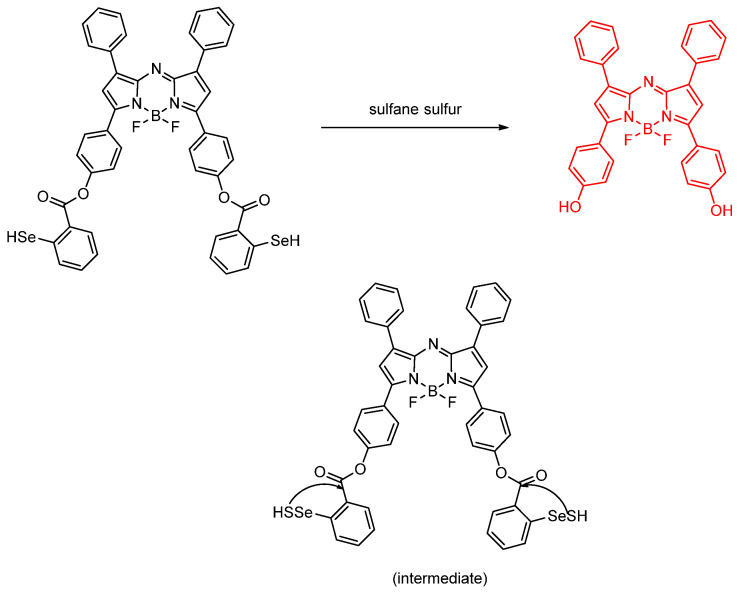
Near–infrared (NIR) chemosensor based on azo–BODIPY and diphenylselenol (BD–di–SeH) probes by Gao et al. for sulfane sulfur analyte detection.

**Figure 20 molecules-26-00692-f020:**
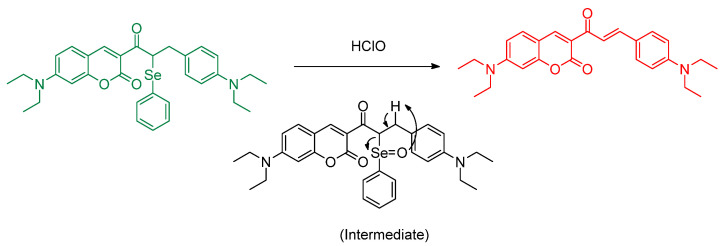
**Coum–Se** chemosensor for HOCl probe reported by Xie et al.

**Figure 21 molecules-26-00692-f021:**
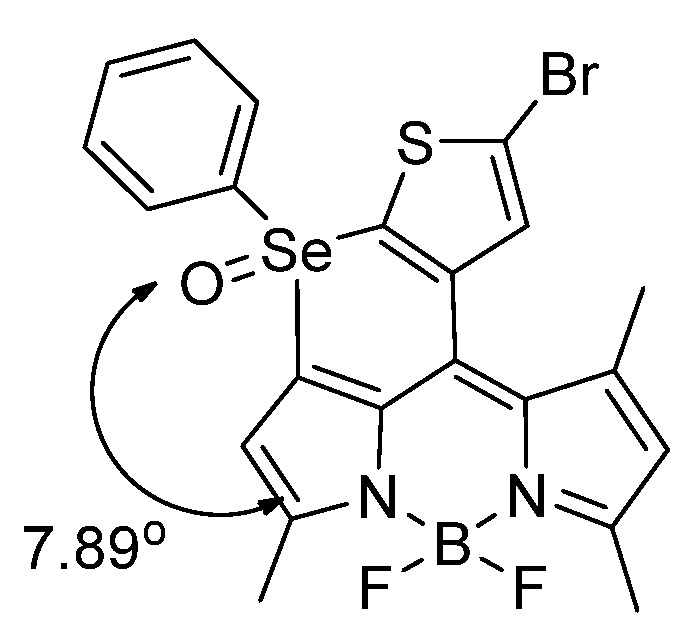
The structure of 5–bromo–p–2–(phenylselanyl)–thiophene–substituted BODIPY and the interfacial angle by Kim et al.

**Figure 22 molecules-26-00692-f022:**
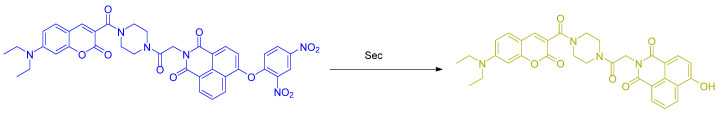
FRET probe **CmNp–Sec** by Zhao et al.

**Figure 23 molecules-26-00692-f023:**
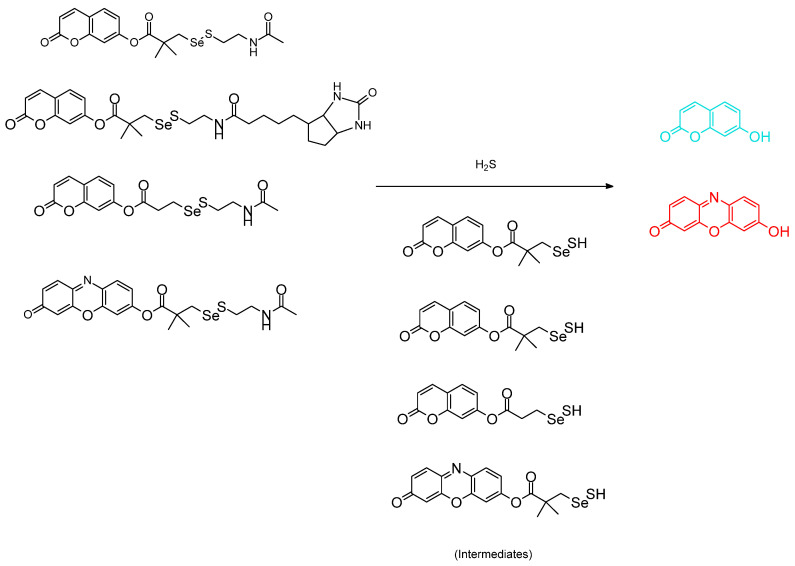
Structures from the report by Suarez et al. describe Selenosulfide gem–Dimethyl Ester conjugation for the goal of improving future H_2_S chemosensing.

**Figure 24 molecules-26-00692-f024:**
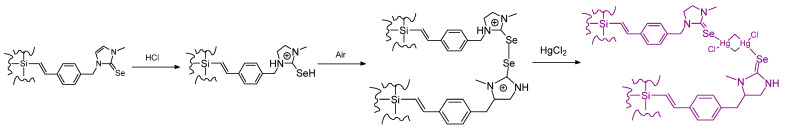
The polyhedral and oligomeric silsequioxane based compound **POSS**–**Se** serving as a chemosensor, by Liu et al.

**Figure 25 molecules-26-00692-f025:**
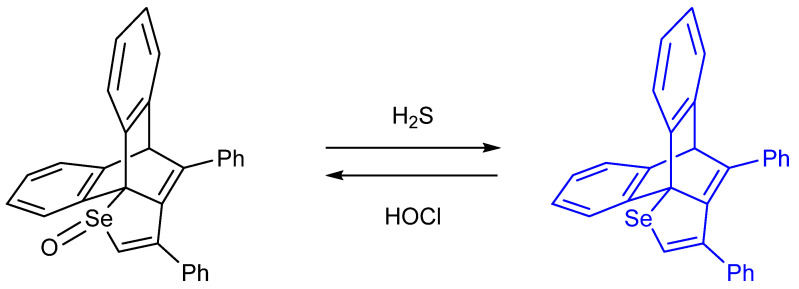
The intriguing dibenzobarrelene structures **DbbSeO** and **MbbSeO** used to detect H_2_S *reversibly/resetably* that involves a ka/selenide redox system by Annaka et al.

**Figure 26 molecules-26-00692-f026:**
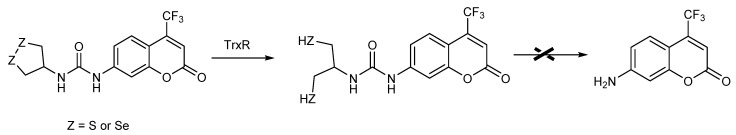
One–step activation of **Fast–TRFS** reported by Li et al.

**Figure 27 molecules-26-00692-f027:**
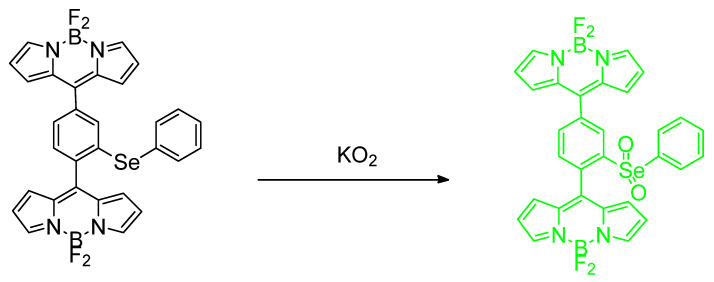
Chemical structure of the tandem fluorophore-substituted phenyl selenide moiety able to undergo oxidation at the Se center by Desmukh et al.

**Figure 28 molecules-26-00692-f028:**
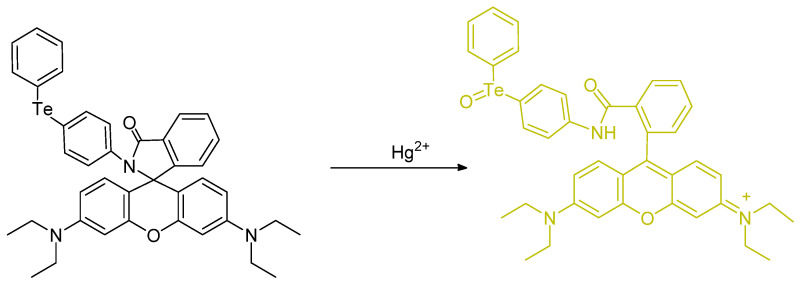
Rhodamine–based phenyl telluride–substituted system used for mercuric detection by Soares–Paulino et al.

**Figure 29 molecules-26-00692-f029:**
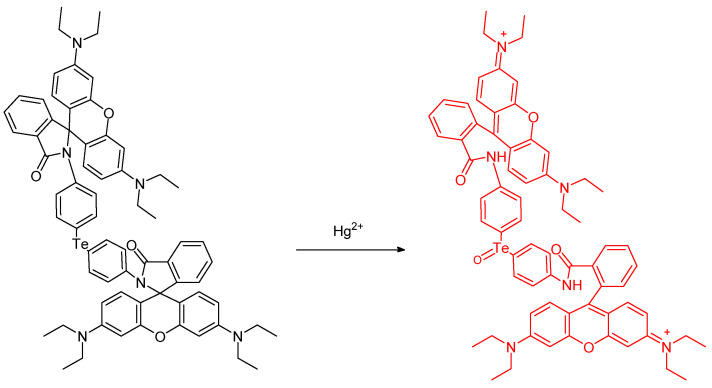
A tellurium–rhodamine B (TRB)–based probe that was studied for Hg^2+^ detection by Soares–Paulino et al.

**Figure 30 molecules-26-00692-f030:**
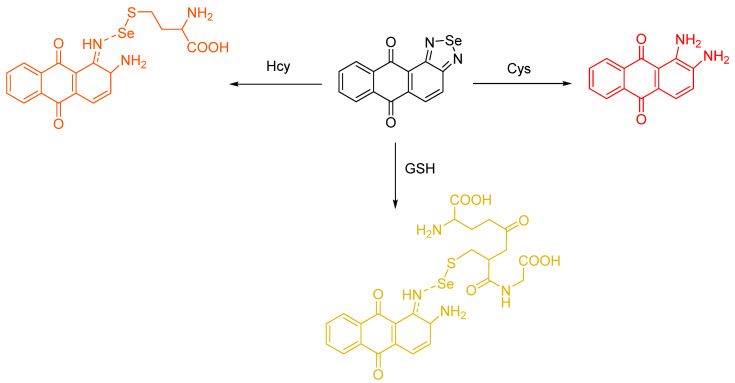
The structures of DAAQ and the proposed mechanism of the probe for differentiating Cys, Hcy, and GSH by Tian et al.

**Figure 31 molecules-26-00692-f031:**
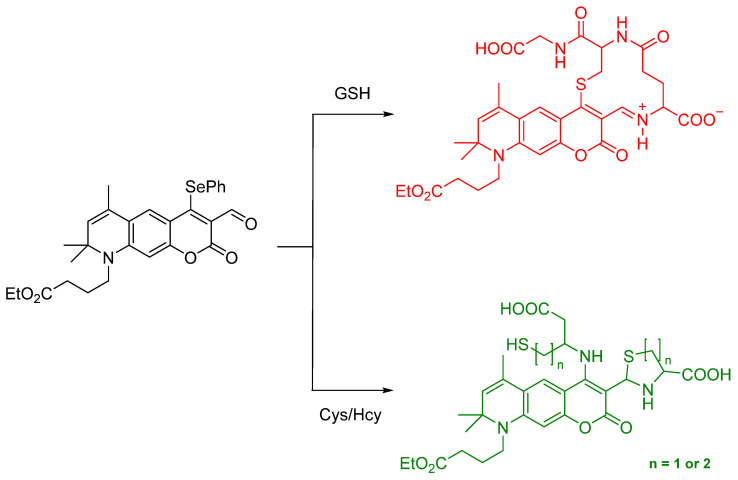
Selective detection of GSH using a coumarin in which the SePh group and CHO group are both serving as adjacent reactive groups.

**Figure 32 molecules-26-00692-f032:**
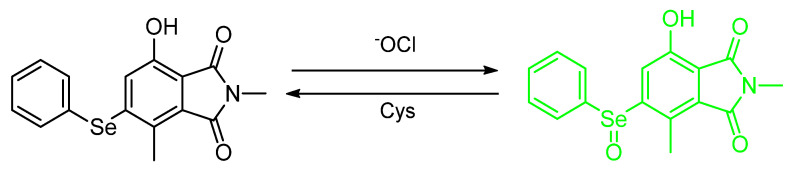
Structure of 7–hydroxy–4–methyl–5–(phenylselanyl)isobenzofuran–1,3–dione probe (**Probe–1**) reported by Mulay et al.

**Figure 33 molecules-26-00692-f033:**
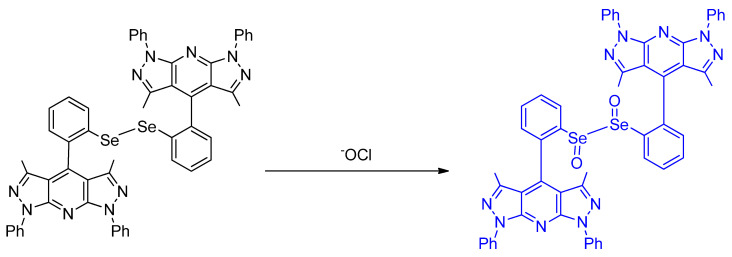
Proposed mechanism of the probe for detection of hypochlorite by Kim et al.

**Figure 34 molecules-26-00692-f034:**
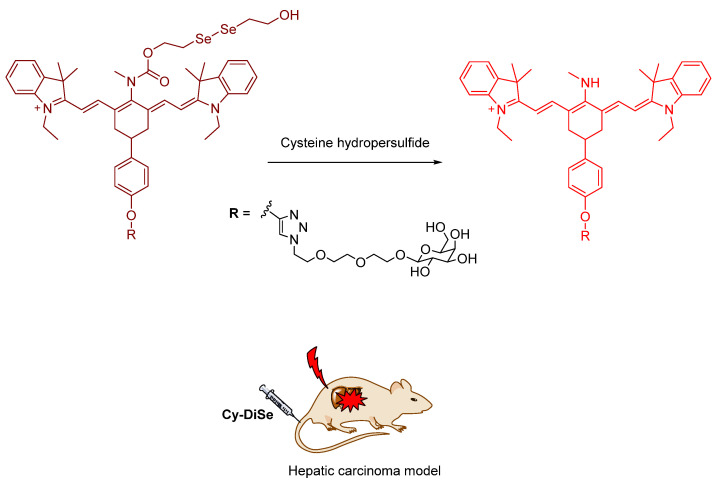
The structure and utility of the **Cy**–**DiSe** probe used in analyte detection by Chen et al.

**Figure 35 molecules-26-00692-f035:**
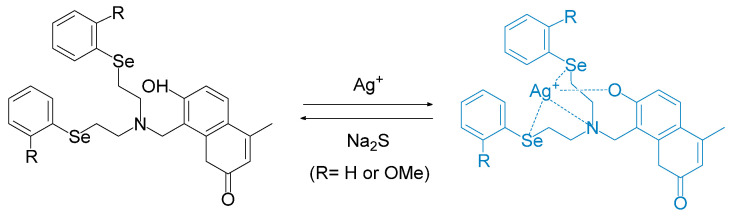
Structures of **SC1** and **SC2** by Su et al. and their reaction with Ag^+^ and Na_2_S.

**Figure 36 molecules-26-00692-f036:**
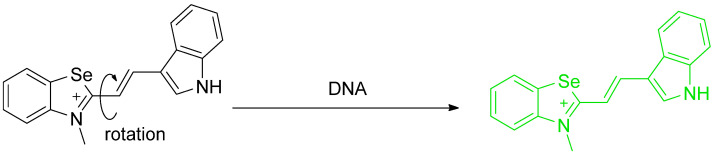
Structure of **PA5** and a schematic of detection mechanism (chemical) reported by Gaur et al.

**Figure 37 molecules-26-00692-f037:**
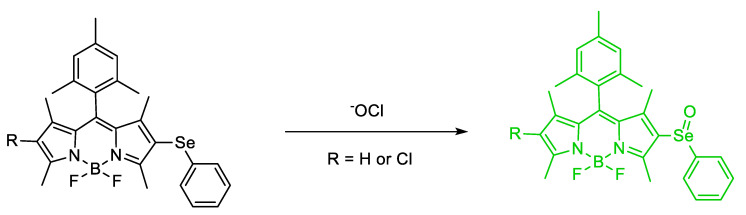
Structures of BODIPY based probes for the detection of ^−^OCl by Mulay et al.

**Figure 38 molecules-26-00692-f038:**
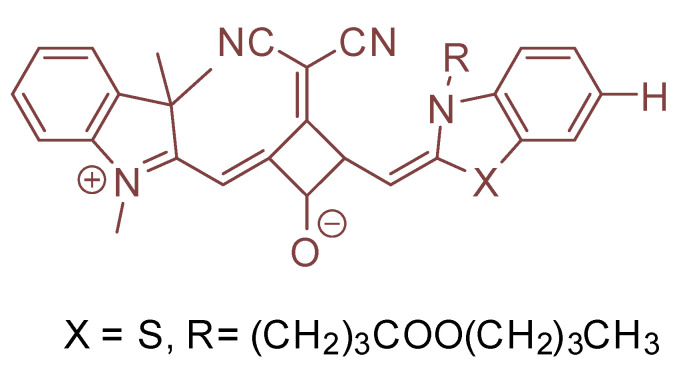
**Compound 7f** squarine compound by Martin et al.

**Figure 39 molecules-26-00692-f039:**
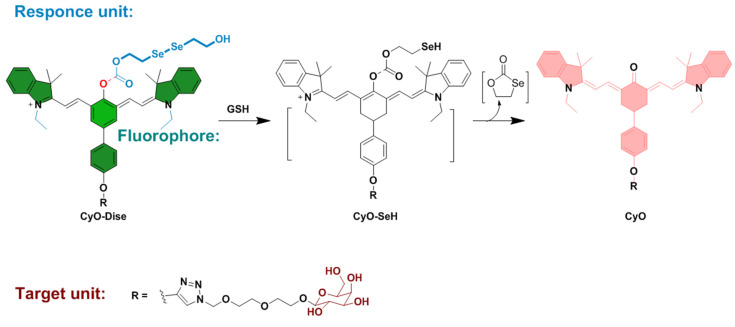
Structure of fluorophore based on hemicyanine synthesized by Han et al.

**Figure 40 molecules-26-00692-f040:**
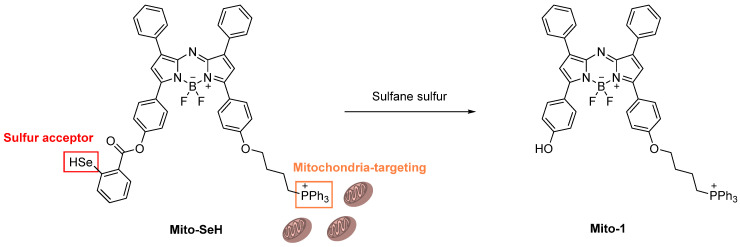
Probe design strategy for sulfane sulfur detection by Gao et al.

**Figure 41 molecules-26-00692-f041:**
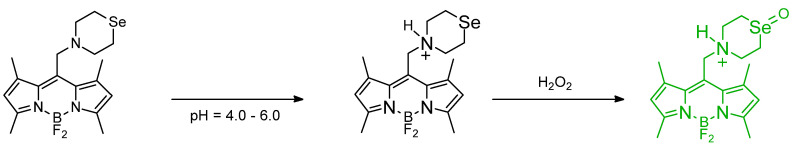
BODIPY system containing an aliphatic Se and protonated amino group synthesized by Xu and Xian.

**Figure 42 molecules-26-00692-f042:**
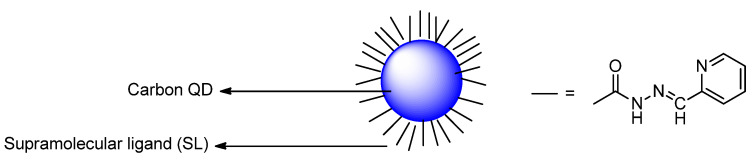
Structure of **CQDs–SL** synthesized by Devi et al.

**Figure 43 molecules-26-00692-f043:**
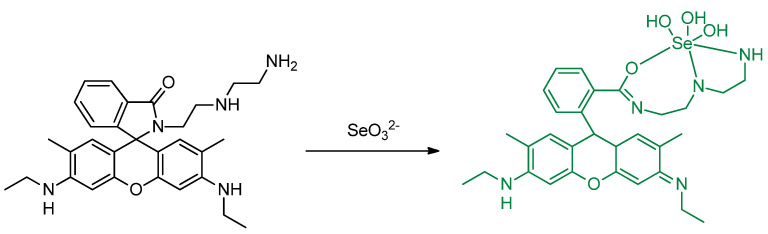
The structure of **ABDO** synthesized by Guan et al. and the binding of selenite analyte to achieve recognition with this probe.

**Figure 44 molecules-26-00692-f044:**
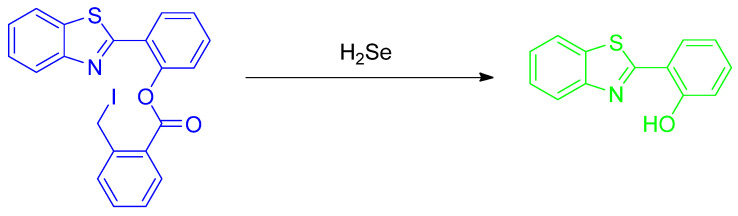
Structure of **HBT**–**Se** and its reaction towards H_2_Se by Xin et al.

**Figure 45 molecules-26-00692-f045:**
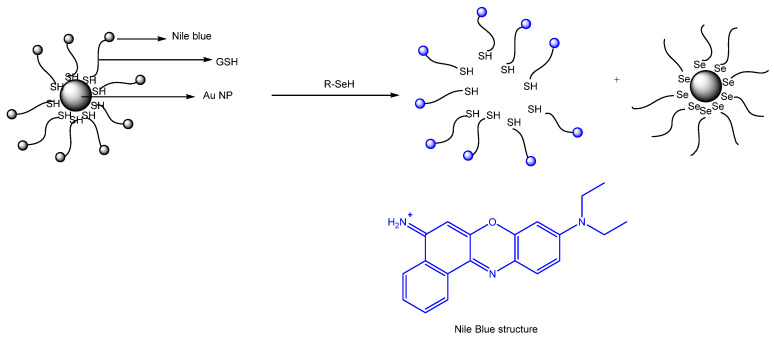
Detection of selenol compound with gold nanoparticles and Nile Blue by Guo et al.

**Figure 46 molecules-26-00692-f046:**
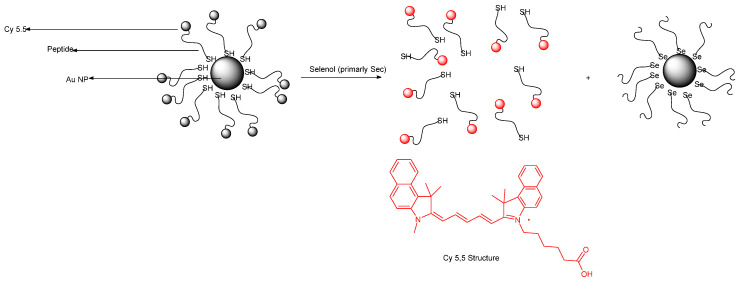
Molecular detection of selenol with gold nanoparticles and Cy–5,5 by Hu et al**.**

**Figure 47 molecules-26-00692-f047:**
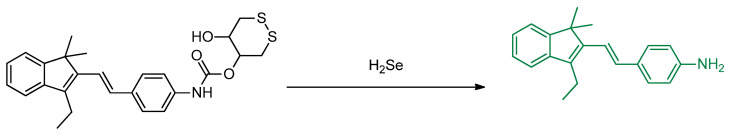
Structure of **Hcy–H_2_Se** and its action of undergoing reaction with and detecting H_2_Se by Kong et al.

**Figure 48 molecules-26-00692-f048:**
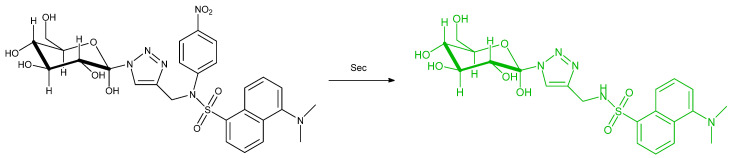
Structure and intended reactivity for chemosensing probe **^NO2^L** studied for selenocysteine detection by Areti et al.

**Figure 49 molecules-26-00692-f049:**
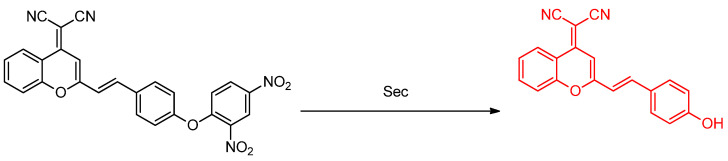
The structure of probe **1** analyzed for selenocysteine based on the 2,4 dinitrophenyl motif by Li et al.

**Figure 50 molecules-26-00692-f050:**
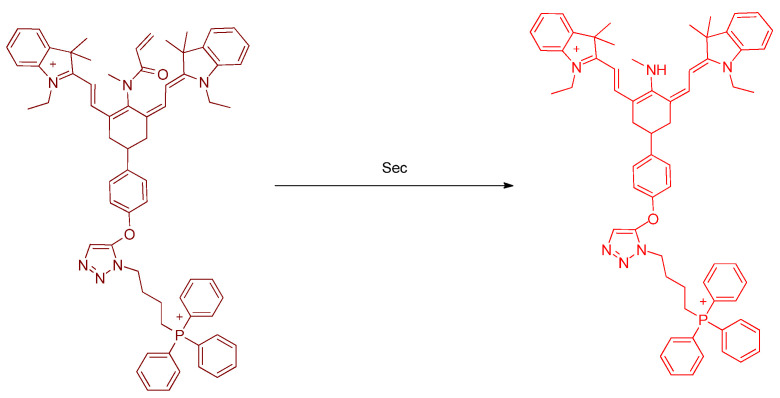
Selenocysteine detection using **Mito–Cy–Sec**, a cyanine based dye by Luo et al.

**Figure 51 molecules-26-00692-f051:**
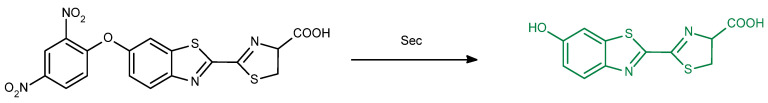
Bioluminescent probes initiated from the 2,4–dinitritro phenyl– group cleavage upon encountering selenocysteine by Zhang et al.

**Figure 52 molecules-26-00692-f052:**
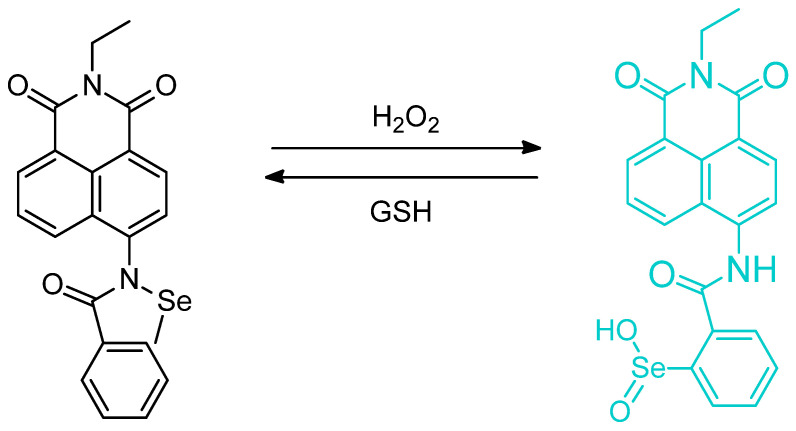
Reaction of **NapEb** by Ungati et al. with H_2_O_2_ and its reversible/resettable reaction with GSH.

**Figure 53 molecules-26-00692-f053:**
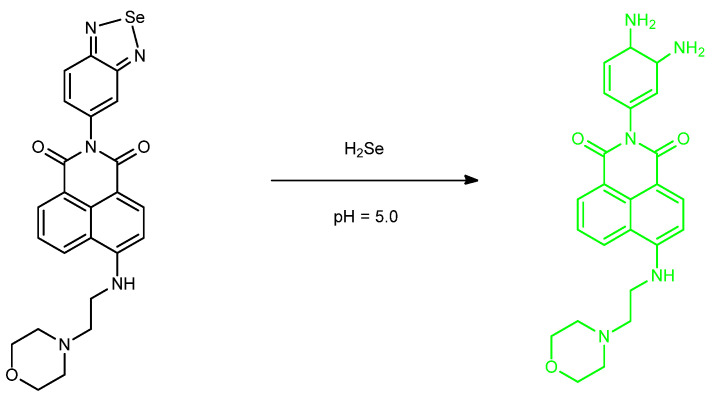
The structure and activity of the chemosensing probe that frees the Se from the *N*,*N*–chelation by Tian et al. The nature of the heterocycle changes dramatically; the native color of the 1,8–naphthalamide is restored.

**Figure 54 molecules-26-00692-f054:**
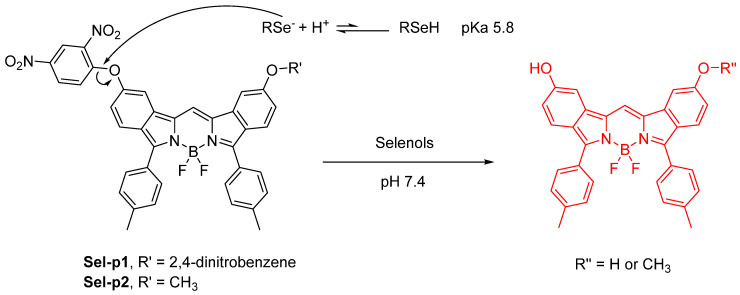
Detection mechanism of **Sel–p1** and **Sel–p2** by Dai et al. towards selenols.

**Figure 55 molecules-26-00692-f055:**
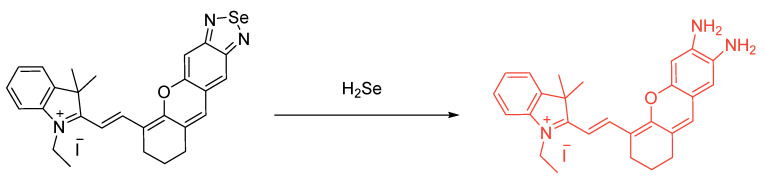
H_2_Se probe **NIR–H_2_Se** by Kong et al. involving the benzene selenodiazole unit.

**Figure 56 molecules-26-00692-f056:**
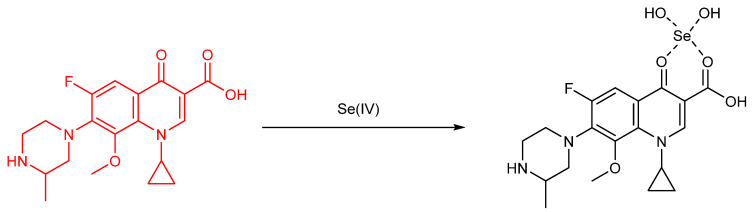
The investigation of **GAT** as a selective Se(IV) fluorescent probe by Liu et al.

**Figure 57 molecules-26-00692-f057:**
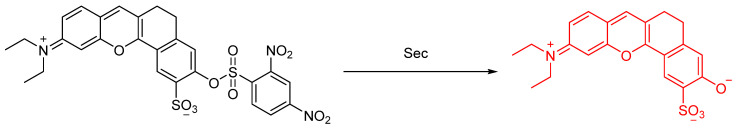
The reaction of the **O–hNRSel** fluorescence probe by Zhang et al. and its selective reaction with selenocysteine.

**Figure 58 molecules-26-00692-f058:**
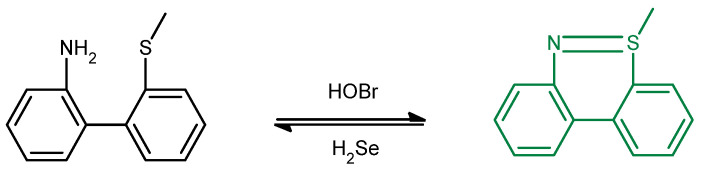
Structure of **BPP**, a simple sulfilimine model system was described by Luan et al.

**Figure 59 molecules-26-00692-f059:**
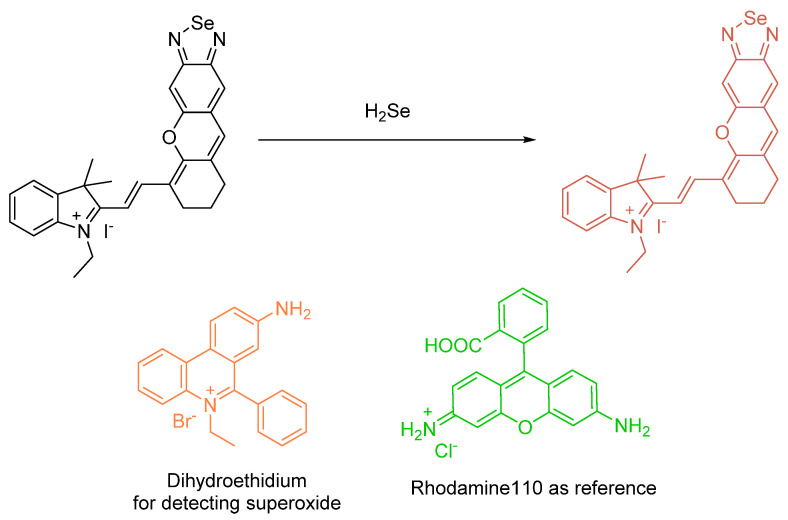
Structure of **NIR–H_2_Se** synthesized by Gao et al. and its reaction with H_2_Se. Structure of dihydroethidium and rhodamine 110 are also depicted.

**Figure 60 molecules-26-00692-f060:**
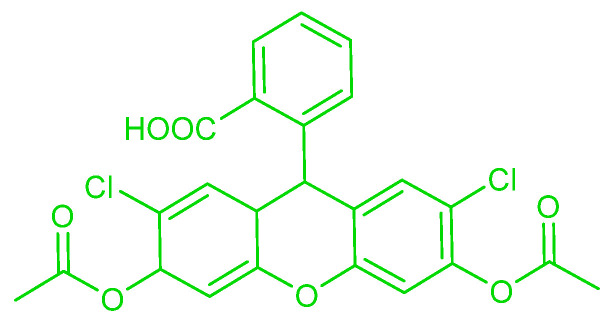
Structure of DCFA–DA by Zwolak et al.

**Figure 61 molecules-26-00692-f061:**
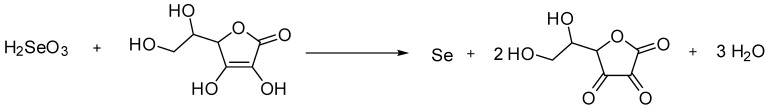
Selenium nanoparticles are formed by reducing them with ascorbic acid with the help of beta–lactoglobulin stabilizer. Investigations of development, physicochemical characterization and cytotoxicity of selenium nanoparticles stabilized by beta–lactoglobulin by Zhang et al.

**Figure 62 molecules-26-00692-f062:**
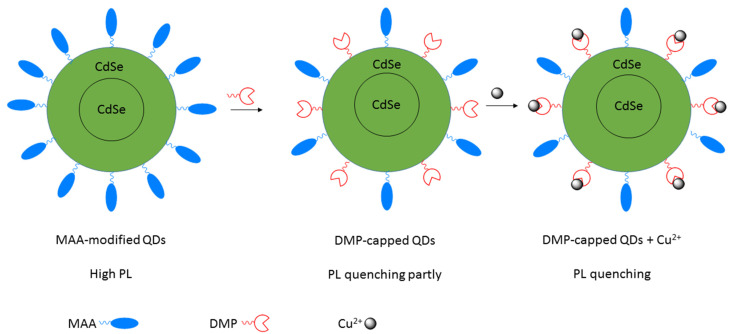
Cu^2+^ detection by functionalized CdSe reported by Zhao et al.

**Figure 63 molecules-26-00692-f063:**
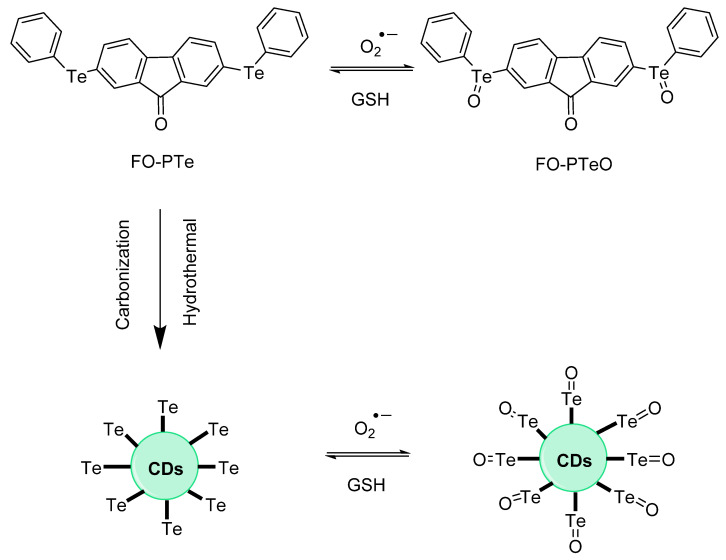
Oxidation of tellurium by superoxide and its reversible reaction with glutathione in carbon dots by Zhang et al.

**Figure 64 molecules-26-00692-f064:**
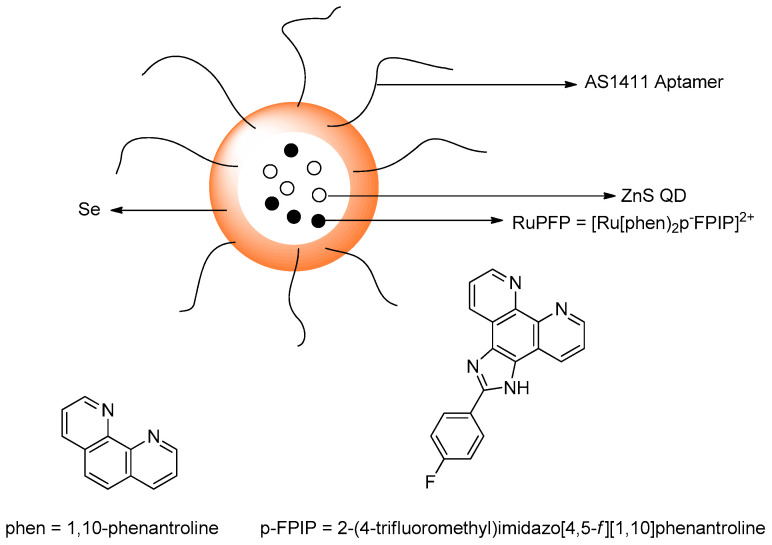
Structure of QDs/Se@Ru(A) by He et al. that is used to antagonize glioblastoma.

**Figure 65 molecules-26-00692-f065:**
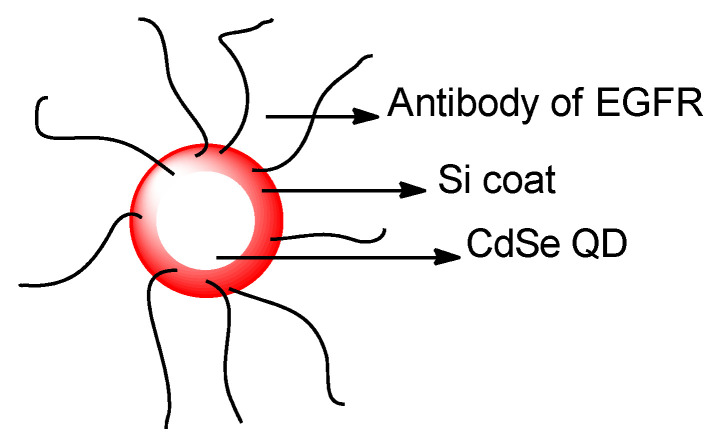
Structure of **QD–Ab** for targeting tumor cell synthesized by Vibin et al.

**Figure 66 molecules-26-00692-f066:**
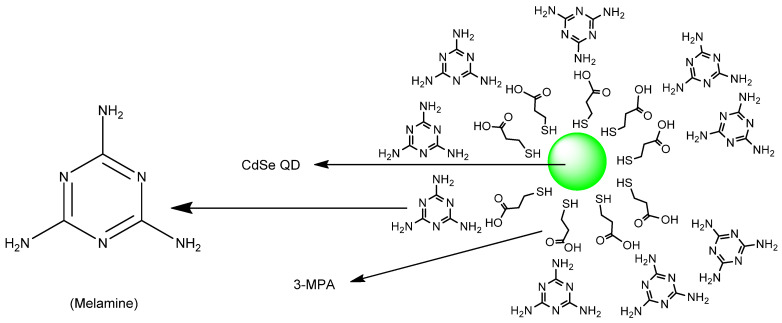
Structure of m–CdSe and its action for the detection of melamine by Singh et al.

**Figure 67 molecules-26-00692-f067:**
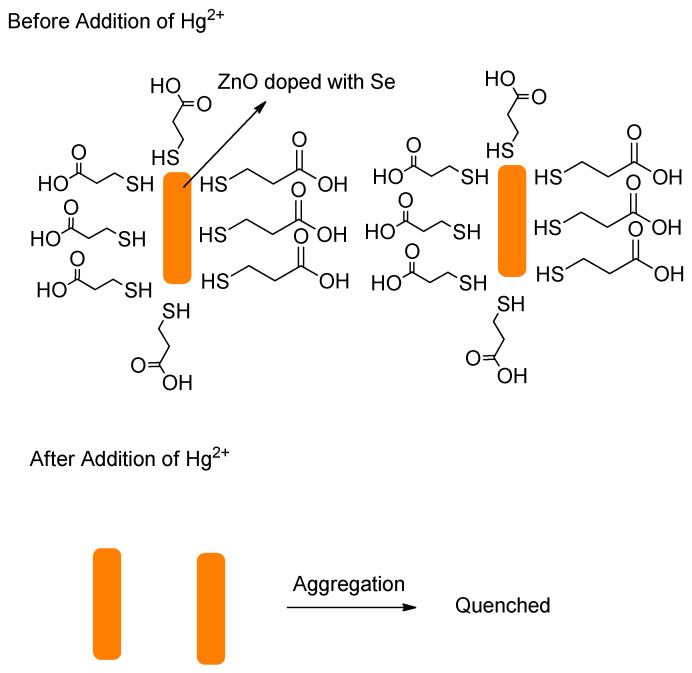
Hg^2+^–based detection from ZnO nanorod doped with selenium and coated with 3–mercaptopropionic acid (3–MPA) by Rao et al.

**Figure 68 molecules-26-00692-f068:**
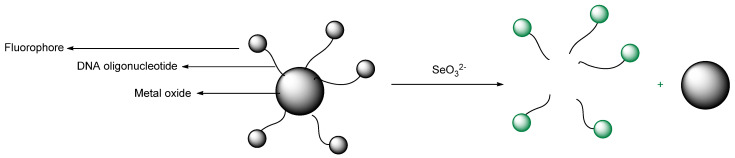
The probing of selenite based on the nanodevice structure and strategy discussed by Yu et al.

**Figure 69 molecules-26-00692-f069:**
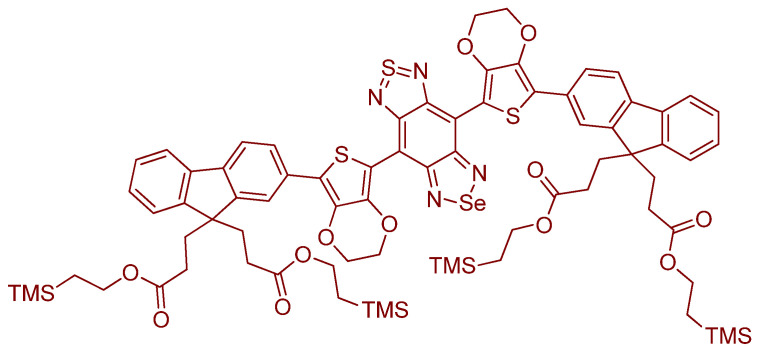
Structure of **SY1080** synthesized by Zhang et al.

**Figure 70 molecules-26-00692-f070:**
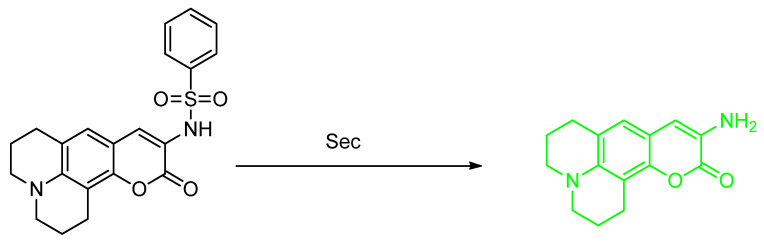
Structure of Probe **3** synthesized by Sun et al.

**Figure 71 molecules-26-00692-f071:**
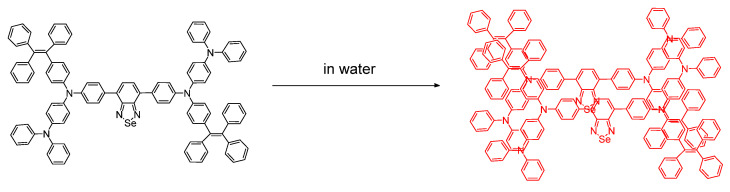
The aggregation-induced fluorescence achieved with a recent probe by Qin et al.

**Figure 72 molecules-26-00692-f072:**
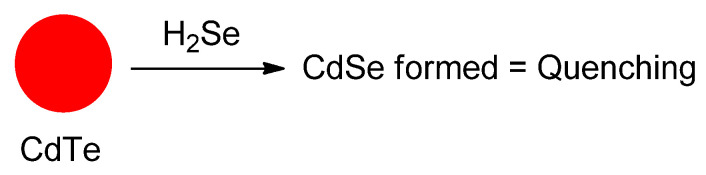
The detection method of H_2_Se using CdTe quantum dots reported by Huang et al.

**Figure 73 molecules-26-00692-f073:**
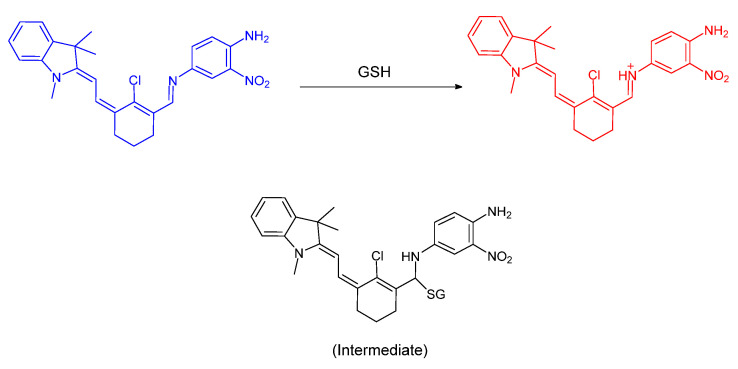
Structure of the GSH- selective probe **HGc** reported by Wang.

**Figure 74 molecules-26-00692-f074:**
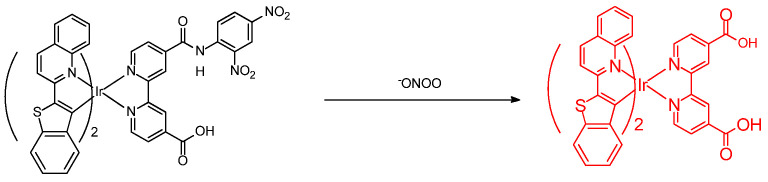
Cleavage by peroxynitrite by Fan, Wong, Wu et al.

**Figure 75 molecules-26-00692-f075:**
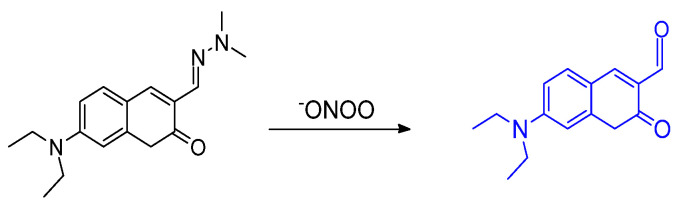
Hydrazone cleavage upon peroxynitrite addition.

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
