# Peer review of "Discussions of Fluorescence in Selenium Chemistry: Recently Reported Probes, Particles, and a Clearer Biological Knowledge†"

_molecules, 2021, doi:10.3390/molecules26030692_

Round 1

Reviewer 1 Report

Extensive English proof read is needed.

Authors use a trivial language that needs to be converted into a scientific one. As example, the first 4 lines of introduction, authors repeat "in the past 5 years" 3 times. Some sentences are senseless such as:

"Important will be that which is not yet mentioned in cell biochemistry but which will likely emerge later"  

"As we prepare this review, a review mainly pertaining on selenium probes, we feel that these papers are setting a foundation for future answers"

"This work is very welcome; it reports selenium as a sensor and biomolecules are important"

Author Response

Some corrections have been made regarding language.

Trivial language and senseless sentences have been addressed and rewritten.

Reviewer 2 Report

The authors have presented an interesting overview on the discussion of fluorescence in selenium chemistry. The review covers the last 5 years and provides useful information on selected selenide and related chemistry. The manuscript provides a contribution to the scientific literature and can be stimulating for a wide range of professional readers  such as chemists, physicists, medicinal chemists and biologists. In my opinion  the paper can be suitable for publication on Molecules although the manuscript is not always easily readable. The schemes should be controlled and uniformed.  

For typing errors:

- the NO2 group isn't totally inside the figure 3.

- Figure 3 is reported twice in the line 200 and 201

-in line 1081 change “Zhang et al, 2018” with “ [117] “and  “Zhang et al., 2018” with “ [119]”

- In line 1082 ref [130] is reported twice

-in line 1097 change “ofthe” with “of the”

-in line 1336 is better to report “ [145]” respect “Zhang et al. 2019”

-in the Figure 69 there isn’t title

-in lines 1510 and 1638  is better to report “ [155]” respect “Liu et al. 2019”

- put to subscript: H2S in  line 1784, Cu2+ in line 1809, Hg2+ in line 1845, H2S in line 1865, Hg2+ in line 1868, H2Se in line 1976, H2Se in lines 1984 and  1986, Na2SeO3 in line 1987, CatiO3 in line 2003, Al2O3 in line 2118 – in line 2119 write in italicus “Fuel”

Author Response

Figure 3 and several figures with the same problem are now revised.

Figure 3 in the later line means figure 3 in the article of Meng et al., not in Figure 3 in our article.

These references are now corrected.

One of the duplicated reports has now been deleted.

The problems relating to “space-placing” are corrected.

References have now been corrected.

The figure caption has now been added.

Author names are added to some paragraphs.

Reviewer 3 Report

In this manuscript, the authors reviewed the literature that focus on selenium species. This article is interesting and valuable, and it is meaningful for researchers that focus on the study of selenium. Several issues about this work are provided below and should be considered again after revision.

  1. Please try adding a table of contents to the front of the article to make it easier for readers to read.
  2. Bad numbering, for example in section 4, only 4.1 was shown. Please clear your mind and number them reasonably.
  3. Do not begin a paragraph in part 9 with punctuation, as this expression is lax.
  4. Please improve the resolution of the image, such as Figure 72. Please bold the chemical bonds.
  5. What are the advantages of selenium in different technologies?
  6. For the section of “Future outlook and suggestions”, advice should be given on each aspect rather than on the literature
  7. The abstract and conclusions were not concise enough and the logical structure was inappropriate, which should be rewritten. The conclusion is poorly written and should outline the contribution of the review.
  8. Some figures needed inspection, such as Figure 8, 10, 17.

Author Response

A table of contents was disallowed upon our initial submission.

These comments have now been addressed and the manuscript has been duly revised.

We wanted to give some specific examples of how research can be extended using specific examples.

The Molecules special issue relates to disulfide and diselenide research and this review was meant to emphasize fluorescent molecules that are attached closely to this topic. We were not going to compare all possible modes of fluorescence with other element centers.

We have now revised both the abstract and conclusion.

Images with bad resolution have now been improved or changed.

The “crowded” structures of fluorescence with AIE mechanism is intentionally stacked to show aggregation.

Round 2

Reviewer 1 Report

Authors made a great effort in making the review properly readable.

I highlight few more criticism just in some graphical parts of their manuscript. Indeed it looks like the figures and schemes were not made by a single person, as a result they are highly inhomogeneous. Authors should put efforts in homogenizing their graphics. Moreover 

Figure 4, structures are weird, please redraw

Figure 11, the chloride is hidden behind the carbonyl

Figure 15, 16, 22, 28 and 41 green and yellow structures are not properly readable, they are too diaphanous

Figure 34, what is CysSSH?

Figure 35, structures are weird, please redraw

Figure 37, I suggest to use R=Cl

Figure 38, Y=H should be removed

Figure 48, there is a dashed bond that needs to be converted into a straight one

Figure 53, check the subscripted numbers, they look weird

Figure 66, elements symbols are oversized compared to the bond length

Figure 73, a cationic nitrogen species is with no reason

Figure 75, What is 1601, 1602 and 1603

Author Response

Dear Reviewers and Editor,

We have hereby revised the manuscript according to the comments of the reviewer. We have made effort to increase the homogeneity and visibility of the figures. The entire manuscript was also edited again for typos. We hope that the manuscript is now found to be in acceptable condition.

(i) Refined: Figure 1, 9, 20, 21, 23, 24, 25, 26, 29, 30, 31, 34, 36, 45, 51, 54, 58, and 59.

(ii) Redrawn: Figure 4, 7, 11, 17, 35, and 38.

(iii) Color change: Figure 6, 7, 12, 13, 15, 16, 22, 28, 41, 52, and 56.

(iv) Authorship order change: Donghyeon Kim (4th → 3rd), Dooronbek Mametov (3rd 4th).

(v) Typos:

well known well-known (line 92)

writtem written (line 134),

reviewdescribing review describing (line 144),

effecting affecting (line 195),

fluorphore fluorophore (line 296),

delocaized delocalized (line 296),

squarine squaraine (line 718),

hrlping helping (line 732),

were was (line 797),

tumeral tumoral (line 918),

anddiscussed and discussed (line 939),

epidemal epidermal (line 1111),

effected affected (line 1196),

fluoresence fluorescence (line 1256),

a an (line 1275),

water soluble water-soluble (line 1352),

indreact indirect (line 1413),

additonal additional (line 1435),

iridum iridium (line 1588),

selenium based selenium-based (line 1604),

identification identifications (line 1270).